# Effectiveness of mental health interventions for older adults in South Asia: A scoping review

Hoimonty Mazumder[1]*, Farah Faizah[1], Easter Protiva Gain[1], Irfath Sharmin Eva[2], Kaniz Ferdouse Mou[1], Nobonita Saha[3], Farzana Rahman[4], Jyoti Das[5], A. M. Khairul Islam[6], Fazilatun Nesa[1], M. Mahbub Hossain[1]

1 Division of Global Health, Research Initiative for Health Equity (RiHE), Khulna, Bangladesh, 2 Response Plan, United Nations Children Fund (UNICEF) Bangladesh, Ukhiya, Bangladesh, 3 Institute of Nutrition and Food Sciences, University of Dhaka, Dhaka, Bangladesh, 4 Department of Biochemistry, Armed Forces Medical College, Dhaka, Bangladesh, 5 Department of Public Health, North South University, Dhaka, Bangladesh, 6 Division of Nutrition and Clinical Service, International Centre for Diarrhoeal Disease Research, Bangladesh (icddr,b), Dhaka, Bangladesh

* hmzumderus@gmail.com

**Data Availability Statement:** All data associated with this paper are retrieved from published materials available in the referred journal articles and has been presented as a part of this review.

## Abstract

### Objective

Mental health problems among older adults are becoming a growing public health concern in South Asia due to continued changes in population dynamics caused by declining fertility rates and increasing life expectancy. This scoping review aimed to explore and summarize evidence about mental health interventions and their impacts on geriatric mental health and highlight gaps and areas for future research.

### Methods

We searched six electronic databases and additional sources for experimental/non-experimental studies evaluating the effectiveness of geriatric mental health interventions in eight countries in the South Asia region from the date of inception of each database up to August 5, 2022. Following the preliminary screening, we extracted data from the eligible articles using a Microsoft Excel data extraction worksheet. We followed Joanna Briggs Institute (JBI) guidelines for this scoping review and reported evidence adhering to the Preferred Reporting Items for Systematic Reviews and Meta-analyses extension for Scoping Reviews (PRISMA-ScR) checklist.

### Results

From a total of 3432 potential articles retrieved, 19 were included in this review following pre-determined eligibility criteria. Across studies, mental health interventions can be broadly categorized into the following types– 1) traditional Yoga, Tai chi, or other meditative movements; 2) behavioral, occupational, or learning-based interventions; 3) tech-based interventions; 4) music therapy; and 5) new healthcare model. The evidence was predominantly based on India (n = 16), whereas three articles were identified from Pakistan. No article was

**Funding:** The authors received no specific funding for this work.

**Competing interests:** The authors have declared that no competing interests exist.

found from six other South Asian countries. Depression and anxiety were the most frequent mental health outcomes, followed by quality of life, cognitive function, self-esteem, physical performance, and many more.

## Conclusion

Although limited, this review found various interventions that have varying effects on different geriatric mental health outcomes. A handful of evidence on mental health intervention in South Asia indicates a lack of acknowledgment that may develop a serious paucity of geriatric mental health practice. Therefore, future researchers are encouraged to conduct empirical studies to understand disease burden, including associated factors of geriatric mental health, which may help to construct contextually appropriate mental health interventions in this region.

## Introduction

Population aging has been evolving worldwide unprecedently since the 20th century [1]. Older adults contributed to 962 million of the world's population in 2017, projected to double by approximately 2.1 billion by 2050 [2]. The growing elderly population is an imminent concern in low-and middle-income countries, including South Asia, which is home to nearly a quarter (23%) of the global population [3]. A drastic drop in fertility rates and increasing life expectancy during the late 20th and 21st centuries have eventually been impacting the age structure of South Asian countries, shifting towards consistently burgeoning group of older adults [1]. As a result, the substantial growth of the elderly population in the South Asia region will take place over the next couple of decades, likely contributing to 334 million of the total population in 2050 [4]. As life expectancy grows, older people frequently encounter various physical, mental, functional, and psychosocial challenges that can seriously affect their quality of life and contribute to a higher family and social burden.

According to the World Health Organization (WHO), developing countries constitute 80% of the global non-communicable disease (NCD) burden [5], where older adults share the most [6]. Mental health among older adults is one of the predominant contributors to the NCD burden. Of adults, approximately 15% aged 60 and over experience a mental disorder attributed to 6.6% of all disabilities globally [7]. However, it is discussed on a limited scale in developing country-context, including South Asia, due to pre-existing stigma, lack of resources, limited healthcare access, and absence of effective health management systems. The prevalence of self-reported depression over 50 years was 47.7%, 40.3%, 40.4%, and 11.4% in Bangladesh, India, Nepal, and Sri Lanka, respectively [8]. A systematic review estimated the prevalence of depression as 34.4% among older people in India [9]. Elderly depressive and anxiety disorders are highly prevalent in Bangladesh, estimated at 55.5% and 55.7%, respectively [10,11]. In Nepal, the prevalence was 15.4% for depression, 18.1% for anxiety, and 12.1% for stress among community-dwelling older adults [12].

While evidence about geriatric mental health disorders in South Asia is highly scarce, continually growing older adults creates a high demand for understanding their complex physical, mental, functional, and other psychosocial problems and associated health and social needs. Mental health deeply encompasses all aspects of human lives, including social, cultural, religious, spiritual, and historical, which may incur stressors of varying extents [13]. Older adults

in South Asia share all these psychosocial stressors, possibly more frequently due to declining capacities, co-morbidities, functional disabilities, frailty, bereavement, financial hardship, and family/social neglect [7]. These stressors can result in social alienation, loneliness, and psychosocial distress among older adults that may eventually lead to various mental and behavioral disorders, thereby impacting the overall geriatric quality of life [7]. Moreover, the intersection of geriatric mental and physical health issues in South Asian countries is critical to address because of the lack of integrated policies, legislation, strategic plans, programs, implementation, and other related measures [3,14]. Higher social stigma, taboos, and lack of awareness are frequently associated with underreporting of geriatric mental illnesses [3]. Also, the lack of skilled human resources affects the timely screening, diagnosis, treatment, and prevention of elderly mental health problems [3,15].

Although pharmacological interventions play a critical role in addressing mental disorders; non-pharmacological or psychosocial interventions such as cognitive behavioral therapy [16], group therapy [17], occupational therapy [18], narrative therapy [19], and culturally appropriate healing practices [20] can be profoundly helpful in improving mental health at the community and population level. Hence, psychosocial interventions refer to a range of interpersonal or informational activities, techniques, or approaches that focus on various aspects such as physical, behavioral, cognitive, emotional, interpersonal, social, or environmental factors in order to improve health outcomes [21]. Many prior research has shown the beneficial impact of psychosocial interventions in fostering post-traumatic growth [22], enhancing the quality of life for people with chronic diseases [23,24], and improving cognitive function, social interaction, and overall well-being among older adults [25–27]. Though addressing geriatric mental disorders requires diverse interventions; however, there is a dearth of empirical and summarized evidence about elderly mental health interventions at the South Asian country or regional level. Also, the epidemiology of geriatric mental disorders is not well-studied, limiting the contextual knowledge and insights about aging mental health burdens, which may affect the development and implementation of appropriate mental health services for older adults.

Moreover, it is necessary for health and social policymakers and practitioners to find high-quality evidence on geriatric mental health interventions for effective policymaking and adopting standard practices in the health system. Hence, socioeconomic and cultural dimensions of growing geriatric mental health problems [28] warrant different psychosocial and community-level interventions that are contextually appropriate to develop an older-friendly environmental and social support network. Therefore, adopting a mental health promotion perspective, this scoping review aimed to examine the nature and extent of existing evidence on non-pharmacological or psychosocial interventions relevant to community geriatric mental health practices in South Asia.

## 2. Materials and methods

### 2.1 Data sources and guidelines

We conducted this scoping review according to the Joanna Briggs Institute (JBI) guidelines for scoping reviews and reported evidence adhering to the Preferred Reporting Items for Systematic Reviews and Meta-Analyses extension for Scoping Reviews (PRISMA-ScR) checklist (S1 Checklist). For scholarly data, we systematically searched Medline, American Psychological Association (APA) PsycInfo, Academic Search Ultimate, Cumulative Index to Nursing and Allied Health Literature (CINAHL), Health Policy Reference Center, and the Web of Science databases from the date of inception of each database up to August 5, 2022, using the specific set of keywords applied with Boolean operators (i.e., "OR," "AND"). The search queries were used across titles, abstracts, subject-specific keywords, and topic fields in respective databases

**Table 1. Search strategy used in this scoping review.**

| Search query | Search keywords applied on titles, abstracts, topics, and subject headings |
|---|---|
| 1 | "older adult*" OR "elder* people" OR "elderly" OR "aging" OR "old* people" OR "geriatric*" OR "gerontolog*" |
| 2 | "mental health" OR "mental wellbeing" OR "mental well-being" OR "mental disorder*" OR "mental illness" OR "psychiatr*" OR "psychological health" OR "psychological distress" OR "psychological impact" OR "psychological outcomes" OR "psychological consequence*" OR "psychological comorbid*" OR "psychosocial problem*" OR "behavioral problem*" OR "behavioral disorder*" OR "cognitive disorder*" OR "cognitive impairment*" OR "emotional distress" OR "depression" OR "depressive disorder*" OR "anxiety" OR "PTSD" OR "PTSS" OR "posttraumatic*" OR "post-traumatic*" OR "addiction" OR "substance use disorders" OR "mood disorder*" OR "affective disorder*" OR "DSM*" OR "psychosis" OR "psychotic" OR "oppositional defiant disorder" OR "hyperactiv*" OR "conduct disorder" OR "obsess*" OR "phobi*" OR "schizophren*" OR "bipolar disorder" OR "anorexia" OR "bulimi*" OR "challenging behav*" |
| 3 | "intervention*" OR "program*" OR "polic*" OR "therap*" OR "counsel*" OR "manag*" OR "cognitive behavio*" |
| 4 | "Afghan*" OR "Bangladesh*" OR "Bhutan*" OR "India*" OR "Maldiv*" OR "Nepal*" OR "Pakistan*" OR "Sri Lanka*" OR "South Asia*" |
| Final search query | 1 AND 2 AND 3 AND 4 |

without using any filter option (Table 1). A complete search strategy for selected databases is provided in the supplementary section (S1 File). The protocol of this review was not registered in any review repository; however, it is available from the authors upon request.

## 2.2 Eligibility criteria

We included articles that met all the following criteria:

a. Original articles reporting mental health interventions (e.g., experimental, non-experimental, quasi-experimental, randomized, pre-post evaluation, etc.),

b. Studies that recruited older participants (age 50 years or over) were included in this review, consistent with previous research involving older adults in this region to make this review more inclusive [29–32]. Hence, we considered the author's perspective of the included studies in defining the aging population,

c. Studies that evaluated psychosocial, behavioral, or non-pharmacological interventions for addressing geriatric mental health problems. It also encompasses interventions if they involve a combination of psychosocial therapies along with any pharmacological treatment, as well as studies that compared the effectiveness of psychosocial interventions with pharmacological medications,

d. Studies that reported at least one mental health outcome, including mental disorders and psychological problems within the scope of the International Classification of Diseases (ICD) or Diagnostic and Statistical Manual (DSM) for mental disorders,

e. Studies that focused on older people living in any of the South Asian countries,

f. Studies that were published in scholarly journals as peer-reviewed articles,

g. The full text of the articles was available in the English language.

This set of comprehensive criteria allowed the potential inclusion of any article that described the effectiveness of mental health interventions for older adults in South Asia.

However, articles not published as peer-reviewed pieces (e.g., editorials, commentaries, pre-prints) or those without intervention (e.g., descriptive papers without outcome evaluation, reviews, case studies) were considered ineligible. Also, citations without full texts or those published in languages other than English were excluded from this review.

We used a cloud-based systematic review management portal (rayaan.ai) to evaluate all retrieved citations. Two reviewers independently reviewed all entries, and a third reviewer assisted them in resolving potential conflicts about eligibility through discussion. All citations eligible for full-text review were examined using the same process and included for data extraction and synthesis.

## 2.3 Data extraction

We used a pre-designed data extraction form that included items on study characteristics (e.g., study design, methods, samples, recruitment strategies), interventions and their contents, and mental health outcomes following the interventions. Three reviewers participated independently in the data extraction process, and a fourth author reviewed all the extracted data to ensure consistency and accuracy. Later, all reviewers participating in data extraction discussed discrepancies and reached a consensus.

## 2.4 Data synthesis

Scoping review provides an evidence map and informs potential gaps by evaluating the collective body of evidence. We narratively summarized the key findings on major study variables, interventions, and all mental health outcomes of interest. These findings were arranged in separate groups and presented in a tabulated form that informed the overview of existing intervention studies and their outcomes.

## 3. Results

Fig 1 illustrates the detailed literature search and selection process adhering to the PRISMA flowchart. The systematic search of electronic databases and other sources yielded 3088 citations. Following deduplication, 1094 citations were included for the preliminary title and abstract screening using pre-determined eligibility criteria. Also, 344 citations were identified and screened from additional sources, including Google Scholar, reference searching, and others. We initially found 26 potential citations for full-text evaluation and finally included 19 articles in this scoping review [31–49].

## 3.1 Characteristics of the included studies

An overview of the included studies is provided in Table 2. The vast majority of studies, 84% (n = 16), were conducted in India [32–45,48,49], and the remaining studies (n = 3) were conducted in Pakistan [31,46,47]. No studies were identified from Afghanistan, Bangladesh, Bhutan, Maldives, Nepal, and Sri Lanka. Most studies (n = 17) were published during 2011–2022. Studies mainly conducted in institutional settings (n = 8) that include old homes, residential homes, and hospices [34,35,38,40,44,45,47,49]; five were conducted in hospital settings [31,36,37,39,41], five in the community [32,33,43,46,48], and the rest recruited samples from both healthcare and community settings [42]. The age range of recruited older adults was 50 years and over, but the mean age of all studies was over 60. Except for one [49], all studies included male and female participants.

Ninety-five percent were intervention studies (n = 18) with sample sizes ranging from 7 to 181. Eleven of them had experimental study design with both intervention and control groups

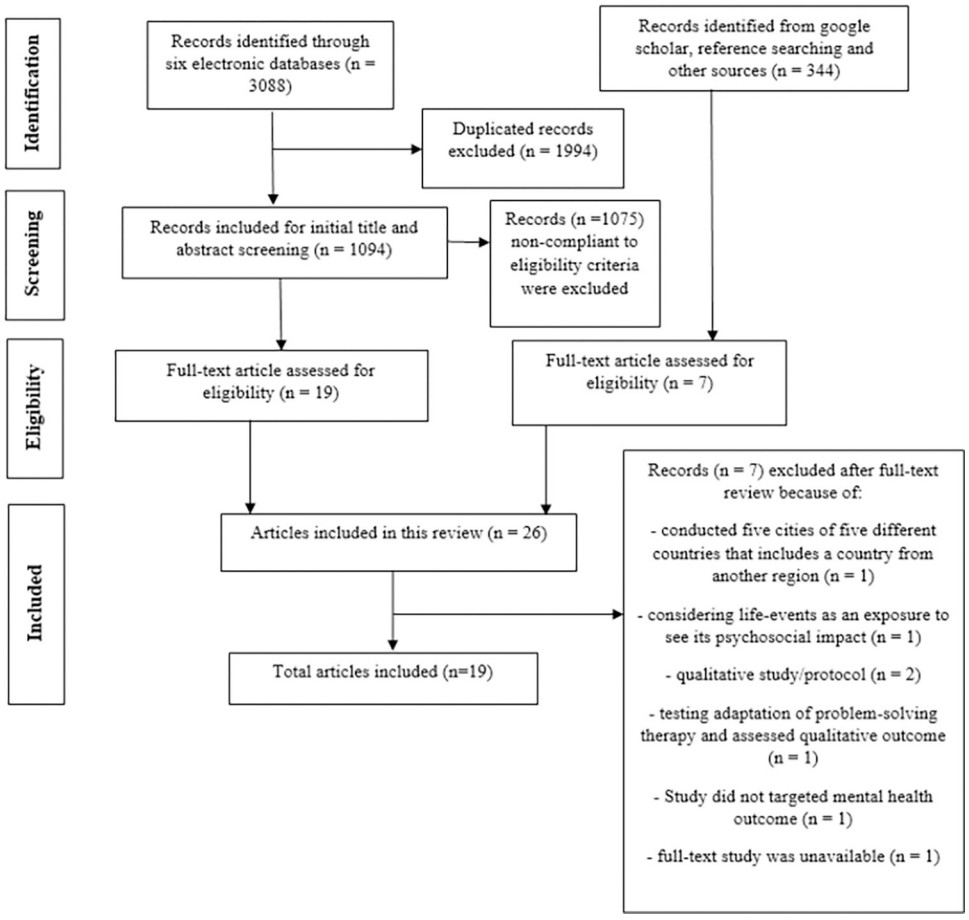

**Fig 1. Flow diagram of the literature retrieval process.**

[32–35,38,39,42–44,48,49], and seven studies followed the pre-post design [31,36,40,41,45–47] applied to the same sample population before and after the intervention. For instance, Dias et al., 2018 randomly assigned lay-provider-led mental health intervention to a group while the control group received usual care [42], whereas Chobe et al., 2022 assigned three yoga and/or ayurveda-based interventions to three different elderly groups with mild cognitive disorders and assessed pre-and post-outcomes of each intervention group [48]. Five intervention studies used randomized techniques to assign participants to the intervention and control arm [34,35,42,43,49]. The only comparative secondary analysis assessed one-year records of mental health access in a newly implemented community-based mental health program compared to a tertiary-facility-based (MCH) mental health service [37]. This secondary analysis assessed the healthcare access of older people in the mental health community outreach clinics compared to a traditional tertiary health-care facility using past one-year health records (216 vs. 187). Over half of the included studies in this review (53%, n = 10) recruited sample populations from a convenient setting, either a hospital or old-age home/residential home/hospice [31,34,36,38–41,45,47,49].

## 3.2 Mental health interventions and their outcomes

Although limited, various interventions evaluated their impact on mental health disorders among South Asian elderlies. Since India is the home of meditative movements and many other traditional practices, most studies have assigned these practices as interventions to

**Table 2. Characteristics of included studies (n = 19).**

| SI | Author, years of publications | Country; Study settings; timeframe/study period | Study design | Recruitment and sampling strategy | Mental health interventions; Comparison characteristics | Sample size (Intervention group and/or comparator); Age (Mean age/ age range); Gender; Participant's characteristics | Outcomes |
|---|---|---|---|---|---|---|---|
| 1 | Dias et al., 2018 [42] | India; Healthcare settings (rural and urban primary healthcare) and the broader community; March 31, 2015- June 2, 2017 | Parallel-grouped randomized controlled trial | Total participants enrolled from rural and urban primary healthcare clinics and randomized by using a computer-generated system into two arms- "Depression in Later Life" (DIL) as the intervention group and "care as usual" (CAU) as the control group; all study personnel, including assessors, were masked to participants' assignment and outcome. | Mental health intervention: Lay counselor-led problem-solving therapy, brief behavioral treatment for insomnia, education if self-care, and assistance in assessing medical and social programs; Comparison group: care as usual. | 181, (DIL Intervention arm: 91, CAU arm: 90); Age range: 60 years and older; Male: 67 (37%), Female: 114 (63%); Older adults with sub-syndrome symptoms of depression | 1) DIL intervention led to a reduced incidence of major depressive episodes compared to the control group (percentage of depression-free participants: 95.1% in the intervention group vs. 87.4% in the control group) (p-value <0.001). 2) This intervention did not affect functional status or cognition among older adults. |
| 2 | Chobe et al., 2022 [48] | India; Community (Urban area); Study continued for 8 weeks. | Three arms-matched control trial | Participants were reached by circulating advertisements in local newspapers and pamphlets in an urban community. Elderly people were screened for eligibility using pre-determined criteria and recruited in this study. | Yoga intervention group received integrated Yoga sessions 6 days a week for a total of eight weeks, AMR intervention group received a polyherbal Ayurvedic formula for 8 weeks; and IY plus AMR group received both techniques. | 72, 25 received Yoga sessions (IY), 23 received Ayurvedic intervention (AMR), and 24 received a combination of both (IY plus AMR); Mean age: IY: 62.40 ± 6.06, AMR: 64.39 ± 7.15, IY plus AMR: 63.21 ± 6.24 (Age range: 55–80 years); Male: 35, Female: 37; Elderly people with mild cognitive impairment. | All three groups significantly improved quality of life, stress, anxiety, and sleep after 8 weeks of intervention (p-value <0.05). Depression was significantly improved in IY plus AMR group. The comparison between the group showed better stress, anxiety, and sleep improvement in the IY plus AMR combination group compared to AMR intervention. |
| 3 | Dev et al., 2014 [40] | India; Institution-based (Two old age homes); Study conducted for 3 weeks. | Experimental study | A total of 40 elderly people with depressive symptoms but who had no cognitive impairment and/ or chronic medical diseases were recruited from two old age homes purposively in a district of Kerala, India. | Music therapy was given half an hour a day for 21 days. This package consisted of old film songs, new film songs, instrumental and classical music based on the raga "Mohanam". Songs were played according to the person's preference. | 40 persons, all of them received music therapy intervention; Age above 60 years; Majority of the participants (60%) were males; Older adults with depressive symptoms. | A significant reduction in the depressive symptoms was observed before and after intervention (p-value: <0.001). After the intervention, the level of depression was effectively reduced in the mild and moderate categories; however, no change was observed in the severe depression category. |

**Table 2.** (Continued)

| SI | Author, years of publications | Country; Study settings; timeframe/study period | Study design | Recruitment and sampling strategy | Mental health interventions; Comparison characteristics | Sample size (Intervention group and/or comparator); Age (Mean age/ age range); Gender; Participant's characteristics | Outcomes |
|---|---|---|---|---|---|---|---|
| 4 | Hariprasad et al., 2013 [36] | India; Hospital setting; Study conducted in 1 month. | Open-label exploratory study to test the feasibility of the validated yoga-based intervention. | 10 elderly people who showed interest in participating in this intervention through workshops and paper advertisements were recruited. | A comprehensive yoga module comprised of warm-up exercises, physical postures, purificatory practices, breathing exercises, and meditation and offered 12 sessions in 1 month. | 10 persons met the pre-determined eligibility criteria, all received the intervention; Mean age: 65.1 years (SD ± 4.8) (Age range: 60–72 years); Male: 10, Female: 2. | All of them could perform meditation techniques without difficulty; some had trouble performing postures; however, most of the older adults found difficulty remembering and completing the entire sessions of yoga-based intervention independently. All individuals reported one or more benefits, such as improved concentration, mood, and backache (p-value <0.05). |
| 5 | Hariprasad et al., 2013 [35] | India; Institution-based (Nine old age homes in and around a city); Study conducted for 6 months. | Single-blind controlled study with block randomization | Interested participants following a lecture given about the study were screened for inclusion based on eligibility criteria, recruited, and then randomized to either yoga or waitlist group by generating a random number table for allocation. | The yoga module comprised loosening exercises, physical postures, breathing exercises, and meditation. A trained therapist offered this program for 60 minutes each session daily for one month. The yoga therapist continued to take 1 weekly session for the next 2 months, and participants were encouraged to self-practice Yoga for the next 3 months. | 87, (yoga intervention = 44, waitlist = 43); Mean age of Intervention group: 75.74 years (±6.46), Mean age of waitlist: 74.78 years (±7.35); 58.1% in yoga group and 62.1% in waitlist group were female; Older adults who met pre-determined eligibility criteria. | Yoga group showed significant improvement (p-value <0.05) in several domains of cognitive function in the elderly, starts from immediate and delayed recall of verbal and visual memory, attention and working memory, verbal fluency, executive function, to processing speed than the waitlist group. |
| 6 | Jacob et al., 2006 [33] | India; Community setting; Study conducted for 3 months. | Community-based intervention study has been implemented for the past 50 years; hence this study followed up two cohorts based on their participation in-built community daycare intervention. | Of 250 interviewed in that community, 41 subjects who belonged to the lowest third on the socioeconomic and social support scale were invited to participate. Twenty elderly people participated in the community intervention, and 21 did not participate. | Intervention group: Community-based daycare center included recreational activities, occupational therapy, counseling services, medical services, and a noon meal; Comparison group: did not attend that program. | 41, (Participated in community intervention = 20, did not participate = 21); Mean age of the Intervention group: 70.3 years (±8.0), and the control group: 70.6 years (±7.5); Male: 8, Female: 33. | Significant reduction (p-value <0.05) in psychiatric morbidity and improvement in quality-of-life scores were observed in the group who attended the program. |

(*Continued*)

**Table 2.** (Continued)

| SI | Author, years of publications | Country; Study settings; timeframe/study period | Study design | Recruitment and sampling strategy | Mental health interventions; Comparison characteristics | Sample size (Intervention group and/or comparator); Age (Mean age/ age range); Gender; Participant's characteristics | Outcomes |
|---|---|---|---|---|---|---|---|
| 7 | Krishnamurthy et al., 2007 [34] | India; Institution-based (old age home); study conducted for a 6-months period. | Three-arms stratified randomized control trial. | Of 90 people from an old age home, 69 participants were stratified according to a 5-year age interval following the screening, then randomized to three groups using a standard random number table. | Yoga group received 75 minutes long yoga sessions daily 6 days a week for 24 weeks that comprised of breathing exercises, loosening exercises, physical postures, voluntarily regulated breathing, yoga-based guided relaxation, and devotional songs; Ayurveda group received herbal preparations twice daily, followed by to drink 200 milliliters of skimmed milk; Control group was not given any intervention. | 50 participants continued study for full length (6 months), Yoga (n = 18), Ayurveda (n = 12) and Wait-list control (n = 20); Mean age of Yoga Group: 70.1 years (SD ± 8.3), Ayurveda group: 72.1 years (SD ± 9.0) and Wait-list Control group: 72.3 years (SD ± 7.4); Included both male and female older participants; Older adults who met pre-determined eligibility criteria. | Only the yoga group showed a significant decrease in geriatric depression following the intervention (p-value <0.001), whereas the Ayurveda and waitlist groups showed no significant difference in geriatric depression after the intervention. |
| 8 | Kumar et al., 2014 [39] | India; Hospital setting; Study period: November 2010 to May 2013. | Open-label randomized control trial | Patients with a Mini-Mental State Examination (MMSE) score between 11 and 23 were recruited from hospitals with the formal education of at least 5 years and the ability to read and understand at least five years of formal schooling, then randomized to intervention and control group using concealment technique. | Intervention group received a novel occupational therapy spanned 70 minutes each session comprising relaxation, physical exercise, personal activities, cognitive exercise, and recreational activities for 5 weeks. In addition, the Control group received regular medication prescribed by the clinician. | 77, (Intervention group: 41, and control group: 36); Mean age of the Intervention group: 69.42 years (age range 60–81), and the mean age of the control group: 69.85 years (age range: 60–83); 80.5% were males, 19.5% were females; Participants with mild to moderate dementia also met other eligibility criteria. | The overall quality of life significantly improved in the experimental group following the intervention (p-value <0.001), whereas in the control group, it declined (p-value = 0.011). Physical and psychological domains in the intervention group increased significantly (p-value <0.001); however, social, and environmental domains of Quality-of-Life scores did not change significantly. This study showed improved physical performance, sleep quality, and energy for daily activities. Moreover, the experimental group observed a greater appreciation of life and a reduction in anxiety and depression. |

*(Continued)*

**Table 2.** (Continued)

| SI | Author, years of publications | Country; Study settings; timeframe/study period | Study design | Recruitment and sampling strategy | Mental health interventions; Comparison characteristics | Sample size (Intervention group and/or comparator); Age (Mean age/ age range); Gender; Participant's characteristics | Outcomes |
|---|---|---|---|---|---|---|---|
| 9 | Kumar et al., 2014 [38] | India; Institution-based (residential home); Study conducted for 5 weeks. | Quasi-experimental study | A total of 60 elderly persons were recruited from different residential homes in a city in India using a non-probability sampling technique. Samples in the experimental and control group were assigned by non-random assignment. | Experimental group received an intervention, namely mindfulness-based stress reduction (MBSR), comprised of observation of breathing, body scan, mindfulness of sound, and thoughts and feelings. This 5-week program has sessions each spanned for 20–30 minutes for 5 days a week; the Control group did not receive any intervention. | 60, Intervention group: 30 and control group: 30; Age range: 60 and above; Males were 53.33% in the experimental group and 60% in the control group. | Significant reduction in depression and increased mindfulness were observed among elderly people in the experimental group after MBSR therapy (p-value <0.001). |
| 10 | Ramanathan et al., 2017 [49] | India; Institution-based (hospice for the aged); Study period: 12 weeks | Randomized control trial | Of participants who expressed interest in this study, 40 females were recruited and randomly assigned to the experimental and control groups using block randomization. | Experimental group received a 12 weeklong yoga therapy comprised of a warm-up, breath-body movement coordination practices, static stretching postures, breathing techniques, and relaxation; the control group received no intervention and continued their regular work at the hospice. | 40 (yoga group: 20 and control group: 20); Mean age of the experimental group: 68.90 ± 7.55 years and the mean age of the control group: 68.20 ± 8.78 years; Only female older adults were recruited; Participants without receiving any specific medical treatment for either depression or anxiety. | Significant improvement was observed in the scores, indicating decreased levels of depression and anxiety and increased levels of self-esteem amongst the elderly women in the experimental group compared to the control group (p-value <0.001). |
| 11 | Riaz et al., 2021 [46] | Pakistan; Participants recruited from a city; Study period: 6 months | Longitudinal pilot study to test the usability of Virtual reality (VR)-based environmental enrichment. | Participants were recruited for this study whose age was above 60 years. | Participants were assigned VR-based environmental enrichment on an individual basis through sessions conducted every 2 weeks overall for a period of 6 months. | 7, all participants received VR-based environmental enrichment intervention; Age range: 60 years and older; Male: 2, Female: 5; Participants with mild cognitive impairment or mild dementia assessed by Montreal Cognitive Assessment (MoCA). | Though cognitive function remained unchanged after 6 months of intervention, a trend toward improved mental wellbeing was observed amongst participants (p-value <0.1). Moreover, qualitative observations from participants' caregivers indicated an overall improvement in patients' reminiscence, mood, engagement, and functional skills. |

(*Continued*)

**Table 2.** (Continued)

| SI | Author, years of publications | Country; Study settings; timeframe/study period | Study design | Recruitment and sampling strategy | Mental health interventions; Comparison characteristics | Sample size (Intervention group and/or comparator); Age (Mean age/ age range); Gender; Participant's characteristics | Outcomes |
|---|---|---|---|---|---|---|---|
| 12 | Tharayil et al., 2013 [37] | India; Hospital setting; analyzed data of past year. | Comparative analysis by examining past year records available at these two healthcare settings (Medical College hospital- MCH vs. Community outreach clinics- DMHP) | Using past year record of outdoor patient visit who south mental health care. | Adopted a new health model termed District mental health program (DMHP) as an approach to decentralize mental healthcare at the community level. Following a year of implementation, an analysis was carried out using data of past one-year mental health access in community outreach clinics compared to a traditional tertiary healthcare facility (MCH). | Older adults sought healthcare at Psychiatry department of medical college hospital: 187, Older adults sought care at district mental health program in peripheral health center: 216; Age range: 60 years and older; Females were 47% in the MCH group and 56.5% in the DMHP group. | A significantly higher number of elderly people (16.5%) are attending the clinics of the DMHP compared to the outpatient service of a teaching (MCH) hospital (9.5%). There was considerably a higher proportion of older women who attended DMHP (56.5%) compared to MCH (47%), though the difference is not statistically significant (p-value<0.0001). |
| 13 | Arshad et al., 2021 [31] | Pakistan; Hospital setting; Study conducted for 6 weeks. | Quasi-experimental study | Purposive sampling who met inclusion criteria | The assigned intervention was a brain training game called "Body and Brain exercises," which spanned 30 minutes each session for 5 days a week. | 18, all were assigned for intervention; Mean age: $62.0 \pm 8.49$ years (Age range: $\geq 50$ years); Male 12, Female 6; Participants with mild cognitive impairment | Significant improvement was observed in the post-test score for measurement for neuropsychological tests that represent improving cognitive abilities of older adults with mild cognitive (p-value <0.05). |
| 14 | Chobe et al., 2020 [32] | India; Community setting; Study period: August 2018 to March 2019 | Non-randomized three-arm clinical trial | Upon an announcement in the local newspaper, elderly people who met the inclusion criteria and finally gave consent to participate were recruited and assigned to one of three intervention groups. | An 8-week-long integrated yoga therapy module was administered to Yoga and Yoga plus Ayurveda group, and a ghee-based polyherbal Ayurvedic formulation was administered to Ayurveda and Yoga plus Ayurveda groups. | 72, (Yoga = 25, Yoga plus Ayurveda = 24, Ayurveda = 23); Mean age: $63.3 \pm 6.44$ (Age range over 55–80 years); Male 35, Female 37; Participants having mild cognitive impairment. | Integrated yoga, ayurveda, combined yoga plus ayurveda interventions were equally beneficial in significantly improving cognitive abilities among the older adults with mild cognitive impairment (p-value <0.05). Integrated yoga, ayurveda, and combined yoga and ayurveda interventions significantly improved learning, attention, processing speed, and working memory. |

*(Continued)*

**Table 2.** (Continued)

| SI | Author, years of publications | Country; Study settings; timeframe/study period | Study design | Recruitment and sampling strategy | Mental health interventions; Comparison characteristics | Sample size (Intervention group and/or comparator); Age (Mean age/ age range); Gender; Participant's characteristics | Outcomes |
|---|---|---|---|---|---|---|---|
| 15 | Sanchetee et al., 2017 [41] | India; Hospital setting; Study period: 4 months | Prospective study | Participants were approached at the hospital and recruited based on their consent and compliance with study eligibility criteria. | A module of meditation named Preksha Meditation was assigned and maintained for 4 months under supervision. | 58, all participants received intervention; Age range: 60–79 years; Male 37, Female 21; recruited those who were physically fit and willing to continue meditation training. | Preksha Meditation caused improvement in all domains such as psychological health, physical health, social health, environmental health, and stress level. |
| 16 | Jafree et al., 2021 [47] | Pakistan; Institution-based (Old age home); Study period: 3 months | Quasi-experimental study | All participants of an old age home who had given official approval for this study were approached, and finally, the study enrolled 42 participants. | The intergenerational learning activities comprised of the broader seven thematic groups: a. Stimulate dialogue, b. Language c. History d. Civic awareness and community belongings e. Music and Creativity f. Character building and Ethics and g. Religious and spiritual dialogue. | Of the total participants the study began with, 18 completed the entire period of intervention; Age range: 65–77 years; Male 13, Female 05. | Improvement in sleep (p-value <0.05), life enjoyment (p-value <0.05), and psychological health (p-value = 0.05). In addition, educated participants showed significant improvement in quality of life after the intervention (p-value = 0.048). |
| 17 | Joseph et al., 2021 [44] | India; Institution-based (Old age home); Study period: January 2013 to July 2014 | Quasi-experimental study | Following official approval, two old age homes were selected and randomly assigned for intervention and control group. | Tai chi therapy was assigned to study participants under supervision, and the control group received routine care. | 150, (Intervention: 75, Control: 75); Age range: 60–80 years; Included both male and female participants. Older adults met eligibility criteria. | Tai chi therapy was significantly effective in reducing depression levels (p-value = 0.006) among older adults. In addition, it was effective in decreasing systolic (p-value <0.001) and diastolic blood pressure (p-value = 0.032) and also in pain management (p-value <0.001). |

*(Continued)*

**Table 2.** (Continued)

| SI | Author, years of publications | Country; Study settings; timeframe/study period | Study design | Recruitment and sampling strategy | Mental health interventions; Comparison characteristics | Sample size (Intervention group and/or comparator); Age (Mean age/ age range); Gender; Participant's characteristics | Outcomes |
|---|---|---|---|---|---|---|---|
| 18 | Maheshwari et al., 2021 –[45] | India; Institution-based (old age home); Study period: 1 month | Experimental study (pre-test post test design) | A simple random sampling technique was used to select a sample of 80 elderly participants from 5 conveniently selected old age homes and assessed eligibility. | Psycho-educational intervention comprised 14 sessions where the following topics were included: a. Health education, b. Information on coronavirus, symptoms, measures to improve immunity and management, c. How to maintain daily routine, d. How to stay healthy, pleasant activities and social skills e. sleep hygiene, f. self-esteem, g. Role of physical activity and diet on stress management, h. Diabetics management, i. Pain management, j. Breathing exercise, k. depression-problem solving behavioral activation, and cognitive behavioral perspective on depression, l. Acidity management, m. management of hypertension, n. side effects of self-medication | 60 participants finally completed the intervention; Age range: 60 years and older; 86.6% were males; all participants received the psycho-education intervention. | The psycho-education training was highly effective in improving self-esteem (p-value <0.001) and reducing depression (p-value <0.001), stress (p-value <0.001), and anxiety symptoms (p-value <0.001). |
| 19 | Ganesh et al., 2021 [43] | India; Community setting; Study period: 3 months | Randomized control trial | Participants were contacted through a billboard, flyers, and the primary care physician's referral for yoga-based lifestyle intervention. 96 Participants with no history of yoga practice were recruited following screening and then randomized into yoga and control group. | Intervention group received Yoga for 3 months at a frequency of 3 sessions per week. The yoga module consists of breathing exercises, loosening exercises with chair support, physical postures, breathing techniques, and meditation; the control group receives no intervention. | 81 participants completed the study, (Yoga = 48, control = 33); Age range: 60–75 years; The yoga group had 71% female participants whereas the control group had 54% females; Participants had no history of Yoga past 12 months and had minimum high school education. | Improved constipation and insomnia amongst the intervention group compared to the control (p-value <0.05). |

examine their effectiveness on various geriatric mental disorders. In total, 53% (n = 10) of included studies evaluated yoga, meditation, Tai chi, and mindfulness-based stress reduction as mental health interventions [32,34–36,38,41,43,44,48,49]. Hence, some studies assigned ayurveda/herbal medication as a comparator or applied a combination of yoga and ayurveda intervention. For example, Chobe et al. 2022, a controlled trial, evaluated the efficacy of an 8-weeklong yoga module, ayurveda treatment, and a combination of yoga and ayurveda in three different intervention groups to evaluate their comparative effects on older adults with mild cognitive impairment [48].

Two studies evaluated two different comprehensive packages of behavioral interventions. First, Dias et al., 2018, assessed the effectiveness of a lay counselor-led behavioral and learning-based intervention for older people with depression that comprised of problem-solving therapy, brief behavioral insomnia treatment, self-care education, and assistance in assessing medical and social programs [42]. Another one, Kumar et al., 2014, assigned a 5-week-long novel occupational therapy targeting older adults with mild to moderate dementia consisting of relaxation, physical exercise, personal activities, cognitive exercise, and recreational activities [39].

Among the included articles, two studies evaluated in-built community/health system intervention programs. Jacob et al., 2006 assessed the effectiveness of a community-based daycare center program on geriatric mental health and quality of life; the other one assessed healthcare utilization of a new model designed as decentralized community-based mental health outreach clinics [33]. Very few studies (n = 2) evaluated technology-oriented interventions such as virtual reality (VR) based environmental enrichment and brain training among older people with mental disorders [31,46]. Some studies (n = 2) assigned learning-based interventions [45,47]. Jafree et al., 2021 evaluated intergenerational learning activities consisting of a broad thematic area such as language, history, civic awareness, music, religious and spiritual dialogue, etc. [47]. The other one assigned a psycho-educational intervention comprising several topics related to health education, sleep hygiene, daily routine, pain management, self-esteem, self-medication, and many more [45]. Interestingly, a 21-day-long instrumental and classical music-based therapy was undertaken as an intervention to explore its effect on geriatric depression [40].

The geriatric mental health outcomes across all these interventions were widely varied. Most meditative or yoga-based interventions significantly reduced geriatric depression and anxiety symptoms. In addition, the yoga-based interventions were highly effective in improving stress, mindfulness, and sleep [48]. Both yoga and a combination of yoga and ayurveda intervention improved the cognitive abilities of the elderly with mild cognitive impairment. Several domains of elderly cognitive function, such as learning, attention, recalling memories, processing speed, and working memories, were significantly improved following the intervention of yoga or combined yoga-ayurveda [32]. Moreover, these meditative interventions significantly improved concentration, mood, self-esteem, stress, and many other psychosocial, physical, and environmental spheres of geriatric quality of life.

Music therapy was found to be effective in reducing mild-to-moderate depression among older adults [40]. The behavioral and learning-based interventions significantly improved geriatric quality of life, physical performance, self-esteem, life enjoyment, and reduced mental disorders such as depression, stress, and anxiety [33,39,45]. Another behavioral intervention led by lay counselors reduced the progression to severe depression among older people with mild depressive symptoms [42]. The technology-based experiments were mainly applied to improve cognitive abilities in older adults. In a longitudinal study, Riaz et al., 2021 assigned a virtual reality (VR)- based environmental enrichment technique to older adults with mild cognitive impairment. After 6-months of intervention, no change was observed in geriatric cognitive function, except for a trend toward improved mental well-being [46]. Another quasi-experimental study assigned a brain training game as an intervention for six weeks on older

participants with mild cognitive impairment [31]. This "Body and Brain Exercise" intervention significantly improved the cognitive abilities of older adults. A significantly higher proportion (16.5%) of the elderly sought mental health care at community-based mental health outreach clinics compared to the psychiatry outpatient service in a medical college hospital (9.5%) [37]. This study was implemented to evaluate mental healthcare access at the community-based mental health outreach centers–a flagship government program piloted in a district of Karnataka state in India to decentralize mental health services at the community level.

Intervention studies or implementation research in mental health substantially contribute to identifying contextually appropriate evidence-based approaches for promoting geriatric mental health status. However, despite a growing elderly population, such a paucity in geriatric mental health research in South Asia highlights a research disparity, possibly associated with its less priority at the public health research and policy level, lack of research investment, inadequate institutional capacities, and other health system/ context-specific factors. Also, inadequate evidence regarding the community-based mental healthcare approach indicates a lack of integration of geriatric mental health services with local stakeholders at the community level, which may limit elderly healthcare access, including social support. Moreover, while growing evidence in developed regions informs health benefits of using digital or technology-based geriatric mental health interventions, we found scanty evidence reflecting serious negligence in advancing geriatric mental health practice beyond the traditional healthcare approach.

## 4. Discussion

This scoping review offered an overview of the information available in the literature about the effectiveness of mental health interventions for older adults in South Asia. Most studies were conducted in India and a few in Pakistan; however, no studies were identified in the other six South Asian countries (Afghanistan, Bangladesh, Bhutan, Nepal, Maldives, and Sri Lanka). Yoga, Tai chi, and other meditative movements have historically been practiced in many countries in South and Eastern Asia [50–52]. Hence, included studies predominantly assigned these meditative movements as interventions to explore their effectiveness on geriatric mental disorders. Moreover, some behavioral or learning-based approaches were evaluated on how physical and psychological aspects of geriatric health were impacted following such interventions. A few papers undertook community-based behavioral interventions to improve geriatric mental health and overall quality of life. Additionally, we found scanty articles in South Asia that employed tech-oriented/ digital mental health interventions for older adults.

Exploring and summarizing evidence about the effectiveness of mental health interventions on various geriatric mental disorders in South Asia revealed a wide range of methodological heterogeneity. The majority were intervention studies; some followed the pre-post design, some applied randomized or non-randomized approaches, and others were quasi-experimental. Primary studies recruited in this review were conducted in diverse settings, including communities, health facilities, and institutions (nursing homes/residential homes/hospices); however, most studies enrolled participants conveniently. The current review observed various study differences, including sample size, population characteristics, sampling strategy, and instruments used in measuring geriatric mental illnesses or quality of life. Most studies were conducted on a small scale, while 44% (n = 8) had a sample size of $\leq$ 50. Hence, notably, study variability can be greater even after simple randomization if the study participants are old and small in number. Small sample sizes also can cause less statistical power, leading to misinterpretation of the intervention results [53]. However, several studies lacked methodological explanations regarding sample selection, recruitment, and sampling strategies, which were likely to increase the risk of bias. Additionally, intervention studies with low or non-significant

effectiveness are less likely to suffer from publication bias or file-drawer syndrome. Therefore, any such studies that were not published and indexed in databases were beyond the scope of the review, which could have provided further insights about the effectiveness on mental health outcomes.

Our review also identified a paucity of research on geriatric mental health interventions in this region. The available evidence was primarily hospital or institution-focused traditional or behavioral interventions, while community-based, digital, or tech-based interventions were highly limited. These interventions mainly focused on geriatric mental health outcomes, where associated physical, psychological, socioeconomic, and cultural factors remained largely un-specified. Prior research highlighted a wide range of interventions such as group-based therapy [54], occupational therapy [55], self-guided therapy [56], physical exercise [57], cognitive training [58], digital or tech-based interventions [59], home-based interventions [60–62], lifestyle improvements [63], community-based intervention [64], nutrition [65], intergenerational program [66], social participation [67], horticulture therapy [68], etc. are found to be effective in improving geriatric mental health and overall quality of life.

The rapid technological advancement in medical sciences since the 20th century has created different avenues in preventive, promotive, and curative healthcare [69], continually emerging as a practical approach offering direct services to people with mental disorders worldwide [70–72]. Unlike institutional healthcare, digital or tech-based geriatric mental health services can bring a significant improvement in implementing evidence-based practices by overcoming a geographic or physical barrier [70,73]. It can also improve the process of screening, assessment, treatment, and follow-up of patients with mental disorders [73]. However, a limited number of studies applied tech-based interventions given the geographic and population scope of this review. Hence, brain training significantly improved geriatric cognitive abilities [31], whereas virtual reality-based intervention showed no cognitive improvement except a trend of positive mental well-being [46]. Such varying effectiveness of tech-based mental health interventions to improve geriatric mental health outcomes suggests the need for further studies with large sample sizes and rigorous experimental designs.

Community-based healthcare is considered viable in delivering- preventive and therapeutic health services in South Asia [74]. Hence, community-based mental health interventions can improve healthcare accessibility, social participation, and physical and cognitive function and promote a social support network for older adults [75]. However, it has been evident that mental health care utilization among older people is low [76–78]; hence, assessing geriatric mental health, especially in low- and middle-income countries, including South Asia, becomes critical [79]. Moreover, a lack of contextual evidence regarding the dynamics of elderly mental health, including its determining factors, may introduce inappropriate measures at the policy level and can cause the demands of older adults to be unmet. Hence, it is essential to establish an integrated healthcare approach encompassing associated physical, psychosocial, cognitive, socioeconomic, cultural, and environmental factors of geriatric mental health.

Designing a mental health intervention for a specific group in a particular context warrants more profound knowledge and insights regarding respective cultural, socioeconomic, environmental, and other psychosocial dimensions. These mental health determinants may directly or indirectly impact geriatric physical, behavioral, and psychological health outcomes. Allen and colleagues discussed how cumulative stress obtained throughout life-course in different contexts, associated with social inequalities, serves as a mechanism that can seriously affect individual/ group mental health conditions [80], which may have a trans-generational impact [81]. They have found a poor and socially disadvantaged group of people disproportionately affected by mental disorders [80] than their counterparts. Older people in South Asia frequently experience low income or financial insecurity; therefore, they are likely to encounter

poverty and a relative decline in family support [82] that eventually may impact elderly mental health conditions and quality of life [83]. Likewise, the cultural impact on geriatric mental health is critical in South Asia, especially in light of this region's rampant political and socio-economic transition since the last century. For example, the cumulative impact and early-life stressors caused by post-colonial sociocultural reforms in India and Pakistan [84,85], post-independence socio-political shifting in Bangladesh [86], the impact of internal conflicts, and political regime changes in Nepal and Sri Lanka [87,88] have exposed older adults to an experience of large-scale socio-political transformation that may have different physical and psycho-social consequences.

Furthermore, continually increasing socioeconomic and demographic transition in South Asia led to substantial changes in family structure, lifestyle, and living patterns, which compelled older adults to live independently. Hence, family and community act as a source of informal financial and social support systems for older adults, exposing them to economic hardship and a dearth of social support and care, along with an existing lack of social connectedness and poor quality of living, which may cause additional psychosocial distress. Therefore, rather than focusing on clinical or other temporary mental health interventions, these critical psychosocial concepts need to be examined regarding their impact on mental health outcomes and should be considered in future intervention designs and deliveries. Therefore, this review calls for extensive research on geriatric mental health, both epidemiological and intervention studies, to understand sociocultural determinants and their impacts on elderly psychosocial health may offer profound contextual understanding to design and implement appropriate interventions.

Finally, we identified little published work on mental health interventions regarding their effectiveness on geriatric mental health outcomes, indicating a serious gap in evidence-based mental health practice in South Asia. Most studies explicitly investigated the effect of interventions on different geriatric mental health outcomes; however, limited evidence made it impossible to identify the most effective intervention. Moreover, there were limitations in the several study methodologies that can introduce bias and inaccurate estimation; therefore, we encourage future researchers to adopt more robust methodological approaches in designing intervention studies. The absence of representative studies from six South Asian countries highlighted research disparity and knowledge gap in this region, which may associate with inadequate mental health policies and programs, low emphasis on scientific evidence at the policy level, a lack of research investment, and limited institutional capacities and collaboration.

There are several limitations in this review. Firstly, this review focused on non-pharmacological interventions; however, we recognize the importance of studying pharmacological mental health interventions. Therefore, the findings of this study alone may not be helpful unless pharmacological and other forms of medical treatment are considered, particularly for patients with severe mental illness. Secondly, critical appraisal or risk of bias assessment is not recommended according to the adhering guidelines; therefore, we did not conduct any methodological quality/ risk of bias assessment of included studies. Thirdly, our approach of retrieving literature from leading selected databases facilitated the inclusion of peer-reviewed articles, whereas unpublished interventions and studies from unselected databases remained beyond our scope. Finally, including articles only in English appeared to be a limitation, which may cause a loss of evidence related to geriatric mental health interventions.

## 5. Conclusion

Mental health disparities among older adults are a critical public health concern in South Asia. The continuous growth of the aging population in this region contributes to the higher disease

burden, including co-morbidities and disabilities that may associate with elevating mental disorders. In addition, many other stressors derived from co-existing socioeconomic, cultural, and environmental factors may have a cumulative psychosocial impact on older adults that can cause substantial psychological and behavioral changes. Our scoping review has summarized evidence about the effect of psychosocial, behavioral, or other non-pharmacological interventions on a broad range of geriatric mental health outcomes. However, limited research on geriatric mental health, including psychosocial epidemiology, warrants further research to bring contextual evidence and insights in order to establish integrated, evidence-based mental health services in South Asia.

## Supporting information

**S1 Checklist. PRISMA-ScR checklist.**
(PDF)

**S1 File. Search strategies.**
(DOCX)

## Author Contributions

**Conceptualization:** Hoimonty Mazumder, M. Mahbub Hossain.

**Data curation:** Hoimonty Mazumder, Farah Faizah, Easter Protiva Gain, Irfath Sharmin Eva, A. M. Khairul Islam, Fazilatun Nesa.

**Formal analysis:** Hoimonty Mazumder, Farah Faizah, Farzana Rahman.

**Investigation:** A. M. Khairul Islam.

**Methodology:** Hoimonty Mazumder, Kaniz Ferdouse Mou, Farzana Rahman, Jyoti Das, Fazilatun Nesa, M. Mahbub Hossain.

**Resources:** Nobonita Saha, M. Mahbub Hossain.

**Supervision:** Hoimonty Mazumder, Jyoti Das.

**Validation:** Hoimonty Mazumder, Easter Protiva Gain, Irfath Sharmin Eva.

**Visualization:** Irfath Sharmin Eva, Kaniz Ferdouse Mou, Nobonita Saha.

**Writing – original draft:** Hoimonty Mazumder, Farah Faizah.

**Writing – review & editing:** Hoimonty Mazumder, M. Mahbub Hossain.

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
