## [Editor Report · Decision Letter 0]

17 Jan 2023

PONE-D-22-29794Mental health interventions for older adults in South Asia: A scoping reviewPLOS ONE

Dear Dr. Hoimonty Mazumder 

Thank you for submitting your manuscript to PLOS ONE. After careful consideration, we feel that it has merit but does not fully meet PLOS ONE’s publication criteria as it currently stands. Therefore, we invite you to submit a revised version of the manuscript that addresses the points raised during the review process.

We look forward to receiving your revised manuscript.

Kind regards,

Girish Thunga, M.Pharm, Ph.D

Academic Editor

PLOS ONE

Journal Requirements:

2. "PLOS requires an ORCID iD for the corresponding author in Editorial Manager on papers submitted after December 6th, 2016. Please ensure that you have an ORCID iD and that it is validated in Editorial Manager. To do this, go to ‘Update my Information’ (in the upper left-hand corner of the main menu), and click on the Fetch/Validate link next to the ORCID field. This will take you to the ORCID site and allow you to create a new iD or authenticate a pre-existing iD in Editorial Manager. Please see the following video for instructions on linking an ORCID iD to your Editorial Manager account: " ext-link-type="uri" xlink:type="simple">https://www.youtube.com/watch?v=_xcclfuvtxQ"

3. We note that you have referenced (ie. Bewick et al. [5]) which has currently not yet been accepted for publication. Please remove this from your References and amend this to state in the body of your manuscript: (ie “Bewick et al. [Unpublished]”) as detailed online in our guide for authors

4. "In your Data Availability statement, you have not specified where the minimal data set underlying the results described in your manuscript can be found. PLOS defines a study's minimal data set as the underlying data used to reach the conclusions drawn in the manuscript and any additional data required to replicate the reported study findings in their entirety. All PLOS journals require that the minimal data set be made fully available. For more information about our data policy, please see http://journals.plos.org/plosone/s/data-availability.

We will update your Data Availability statement to reflect the information you provide in your cover letter."

Additional Editor Comments (if provided):

The authors aimed to perform a scoping review to explore the “Mental health interventions for older adults in South Asia: A scoping review” which is relevant and need of the hour. The manuscript looks fine. However, following issues needs to be addressed.

Title:

1. Title looks incomplete: It can be “Impact of MH on older adult……..

Abstract

2. The abstract can be written in a structured format (it is fine if it depends on authors guideline)

3. Line No: 5; This sentence “for observational/experimental studies evaluating the impact on geriatric mental health in eight countries in the South Asia region” looks incomplete

4. Write a proper conclusion by authors from their observations before the conclusion

Introduction

5. Introduction looks fine, written well

Methodology

6. A “Preferred Reporting Items for Systematic reviews and Meta-Analyses extension for Scoping Reviews (PRISMA-ScR) checklist should be submitted as a supplementary file

7. Disclose the information on registration of protocol for this review

8. Keep the reference for the Eligibility criteria; point b) for Old age threshold

9. Methodological Quality/risk of bias assessment of included studies is missing

Results

10. The inclusion criteria specify the studies to be “non-pharmacological” and authors included the Ayurvedic/herbal medicine (or preparations), which can be considered as pharmacological treatment. Please justify or revise accordingly

11. To say this is a scoping review, authors should add a section describing what is the gap and measures to overcome in the results section. Or else this looks like a systematic review

Overall

12. The English Language can be improved further
---

## [Author Response · Author response to Decision Letter 0]

8 Feb 2023

Responses to the Reviewers 

Thank you for your guidance. We have revised and formatted our manuscript using PLOS ONE’s style requirements. 

2. "PLOS requires an ORCID iD for the corresponding author in Editorial Manager on papers submitted after December 6th, 2016. Please ensure that you have an ORCID iD and that it is validated in Editorial Manager. To do this, go to ‘Update my Information’ (in the upper left-hand corner of the main menu), and click on the Fetch/Validate link next to the ORCID field. This will take you to the ORCID site and allow you to create a new iD or authenticate a pre-existing iD in Editorial Manager. Please see the following video for instructions on linking an ORCID iD to your Editorial Manager account: https://www.youtube.com/watch?v=_xcclfuvtxQ"

Thank you for bringing this to our notice. We have linked an ORCID ID of the corresponding author to the Editorial Manager.

3. We note that you have referenced (ie. Bewick et al. [5]) which has currently not yet been accepted for publication. Please remove this from your References and amend this to state in the body of your manuscript: (ie “Bewick et al. [Unpublished]”) as detailed online in our guide for authors

Thank you for this comment. We have reviewed our manuscript and found no unpublished article; however, we found one article where reference needed to be updated. Therefore, we have included all information related to that article’s publication in the reference section [reference 18].

4. "In your Data Availability statement, you have not specified where the minimal data set underlying the results described in your manuscript can be found. PLOS defines a study's minimal data set as the underlying data used to reach the conclusions drawn in the manuscript and any additional data required to replicate the reported study findings in their entirety. All PLOS journals require that the minimal data set be made fully available. For more information about our data policy, please see http://journals.plos.org/plosone/s/data-availability.

Thank you for this comment. All data associated with this manuscript are retrieved from published materials available in the referred journal articles and has been presented as a part of this review. 

Title: 

1. Title looks incomplete: It can be “Impact of MH on older adult……..

Thank you for bringing this to our notice. We have revised our manuscript's title and rephrased some areas throughout the manuscript to keep in line with the study title.

Abstract

2. The abstract can be written in a structured format (it is fine if it depends on authors guideline)

Thanks to the Editor for this important feedback. We have reviewed some scoping reviews published in PLOS ONE and revised the abstract of this manuscript accordingly (track-changed part). 

3. Line No: 5; This sentence “for observational/experimental studies evaluating the impact on geriatric mental health in eight countries in the South Asia region” looks incomplete.

Thank you for this comment. We have rephased this sentence in the revised version. 

4. Write a proper conclusion by authors from their observations before the conclusion.

Thank you for bringing this to our notice. We have added a conclusive statement from our study observation in the conclusion part of the revised abstract. 

Introduction

5. Introduction looks fine, written well. 

Thank you so much. I appreciate this feedback. 

Methodology

6. A “Preferred Reporting Items for Systematic reviews and Meta-Analyses extension for Scoping Reviews (PRISMA-ScR) checklist should be submitted as a supplementary file. 

Thanks to the Editor. I uploaded a copy of PRISMA-ScR checklist in the supplementary file. 

7. Disclose the information on registration of protocol for this review. 

Thank you for this feedback. We disclosed the information regarding protocol registration of this review in the method under the “Data sources and guidelines” sub-section of the methods.

8. Keep the reference for the Eligibility criteria; point b) for Old age threshold

Thank you for this comment. We have rephrased the sentence to bring more clarity and kept the references with it. 

9. Methodological Quality/risk of bias assessment of included studies is missing. 

Thanks to the Editor for this comment. In this review, we did not conduct any methodological quality/ risk of bias assessment of included studies. According to JBI manual, Scoping review can be used to map the available evidence, identify/ clarify key concepts in a given field (1,2). It is useful to undertake as a preliminary exercise or for examining evidence when there is a clear knowledge gap exists (1–3). Since scoping review intends to map existing evidence in a broader extent rather than synthesizing clinically meaningful information, therefore critical appraisal or risk of bias assessment is not recommended in the existing scoping review guidelines (3,4) and we mentioned it in our study limitation (in the last paragraph of the discussion section).

Results

10. The inclusion criteria specify the studies to be “non-pharmacological” and authors included the Ayurvedic/herbal medicine (or preparations), which can be considered as pharmacological treatment. Please justify or revise accordingly.

Thank you for bringing this to our notice. We included any psychosocial, behavioral, or non-pharmacological interventions in this review. We considered it eligible if any psychosocial intervention was applied in combination with pharmacological treatment or assigned as a comparator to see the effectiveness of psychosocial/ behavioral therapies. That’s why we included those interventions which offered yoga or other meditative intervention in combination with ayurvedic medicine. We have revised the eligibility criteria in the method section to reflect this approach with more clarity.

11. To say this is a scoping review, authors should add a section describing what is the gap and measures to overcome in the results section. Or else this looks like a systematic review.

Thank you for bringing it to our notice. According to guidelines, a scoping review map and present existing evidence in the result section and discussion part allows for a summary of the main results, general interpretation, potential limitations, and measures. Therefore, we presented existing evidence in the result section and summarized all evidence, interpretation, limitation, and potential implications/ measures in the discussion section. To address this feedback, we have added a section to briefly discuss the key research gap identified by this review in the result section (last paragraph of the results section).

Overall

12. The English Language can be improved further.

Thank you for this comment. The manuscript has been reviewed by a native English speaker and revised accordingly.

Reference list: 

1. 11.1.1 Why a scoping review? - JBI Manual for Evidence Synthesis - JBI Global Wiki [Internet]. [cited 2023 Jan 24]. Available from: https://jbi-global-wiki.refined.site/space/MANUAL/4687794/11.1.1+Why+a+scoping+review%3F

2. Arksey H, O’Malley L. Scoping studies: towards a methodological framework. Int J Soc Res Methodol [Internet]. 2005 Feb 1 [cited 2023 Jan 24];8(1):19–32. Available from: https://doi.org/10.1080/1364557032000119616

3. Peters MDJ, Godfrey CM, Khalil H, McInerney P, Parker D, Soares CB. Guidance for conducting systematic scoping reviews. JBI Evid Implement [Internet]. 2015 Sep [cited 2023 Jan 24];13(3):141. Available from: https://journals.lww.com/ijebh/FullText/2015/09000/Guidance_for_conducting_systematic_scoping_reviews.5.aspx

4. Peters MDJ, Marnie C, Tricco AC, Pollock D, Munn Z, Alexander L, et al. Updated methodological guidance for the conduct of scoping reviews. JBI Evid Synth. 2020 Oct;18(10):2119–26.

---

## [Decision Letter · Decision Letter 1]

11 Apr 2023

PONE-D-22-29794R1Effectiveness of mental health interventions for older adults in South Asia: A scoping reviewPLOS ONE

Dear Dr. Mazumder,

Thank you for submitting your manuscript to PLOS ONE. After careful consideration, we feel that it has merit but does not fully meet PLOS ONE’s publication criteria as it currently stands. Therefore, we invite you to submit a revised version of the manuscript that addresses the points raised during the review process.

 1. Avoid using nonstandard abbreviations such as "MH".2. "... limited, variousinterventions" - typo here.3. The authors seem to allude to the supposed effectiveness of digital or tech-based interventions and lament at the paucity of research on such interventions in South Asia. If so, what are the data and findings on effectiveness? More elaboration can be provided in the discussion section.

If applicable, we recommend that you deposit your laboratory protocols in protocols.io to enhance the reproducibility of your results. Protocols.io assigns your protocol its own identifier (DOI) so that it can be cited independently in the future. For instructions see: https://journals.plos.org/plosone/s/submission-guidelines#loc-laboratory-protocols. Additionally, PLOS ONE offers an option for publishing peer-reviewed Lab Protocol articles, which describe protocols hosted on protocols.io. Read more information on sharing protocols at https://plos.org/protocols?utm_medium=editorial-emailutm_source=authorlettersutm_campaign=protocols.

We look forward to receiving your revised manuscript.

Kind regards,

Qin Xiang Ng, MD, MPH

Academic Editor

PLOS ONE

Reviewers' comments:

Reviewer's Responses to Questions

**Comments to the Author**

1. If the authors have adequately addressed your comments raised in a previous round of review and you feel that this manuscript is now acceptable for publication, you may indicate that here to bypass the “Comments to the Author” section, enter your conflict of interest statement in the “Confidential to Editor” section, and submit your "Accept" recommendation.

Reviewer #1: All comments have been addressed

Reviewer #2: (No Response)

Reviewer #3: (No Response)

2. Is the manuscript technically sound, and do the data support the conclusions?

Reviewer #1: Yes

Reviewer #2: Yes

Reviewer #3: Partly

3. Has the statistical analysis been performed appropriately and rigorously? 

Reviewer #1: N/A

Reviewer #2: N/A

Reviewer #3: N/A

4. Have the authors made all data underlying the findings in their manuscript fully available?

Reviewer #1: Yes

Reviewer #2: Yes

Reviewer #3: No

5. Is the manuscript presented in an intelligible fashion and written in standard English?

Reviewer #1: Yes

Reviewer #2: Yes

Reviewer #3: Yes

6. Review Comments to the Author

Reviewer #1: Thank you for giving me the oportunity to review this manuscript entitled “Mental Health Interventions for older adults in South Asia: a scoping review”.

In my opinion, it’s an interesting manuscript and opens doors for future research. It’s a relevant and current topic, general fulfilling the steps of a scoping review

As for the manuscript, there are some comments or suggestions for alteration:

1 - Title:

- There seem to be two titles:

(i) "Mental Health Interventions for Older People in South Asia: A Scoping Review";

(ii) "Effectiveness of mental health interventions for older adults in South Asia: A scoping review".

The second title is more complete and better reflects the content of the article.

- The keywords reflect the content; however, I suggest you include "effectiveness" or "outcomes" or "impact" or "health gains"

2 - Introduction:

- It's clear, short and simple.

- The introduction reveals an adequate understanding of the relevant literature in the field and cites an appropriate range of literature sources.

3 - Methodology:

The methods employed are adequate.

However:

- A supplementary file with the PRISMA-ScR should be submitted

- Quality/risk of bias assessment of included studies is missing

4 - Results:

The results are clearly presented and properly analyzed.

However, there is the inclusion of a study with an intervention that can be considered a pharmacological intervention; however, there is an eligibility criterion that excludes studies with pharmacological interventions. I suggest that the results and conclusions clearly relate to the review question, namely the conceptualization and measurement of well-being.

As this is a scope review, I suggest that the gap found and the measures to be overcome be presented more clearly and explicitly.

Reviewer #2: Please see file attached.

Reviewer #3: First of all, it is always challenging, for both reviewer and authors, when a second reviewer review an already revised draft of a manuscript. The authors will have to deal with two individuals’ preferences, standards, and hang-ups. For instance, I may comment on something that has been a response to the previous reviewer’s comments.

This scoping review aims to identify mental health interventions conducted and published in countries that represent South Asia: Afghanistan, Bangladesh, Bhutan, India, Maldives, Nepal, Pakistan, and Sri Lanka. From 3432 citations, the authors selected 19, of which 16 were published in India. All publications should have been published in English to be included in the review. For a large part the studies had small sample sizes and covered a wide range of formal or informal interventions, with wide methodological variation in terms of inclusion criteria and population characteristics, and to little degree comparable

General comments

First of all, the manuscript is well written, and English is mostly excellent in my opinion.

I would have preferred a less «wordy» approach. To me some sequences, both in the introduction and the discussion sections appear as the authors is rattling off a list of examples, that not necessarily need to be listed. For instance the authors use the «and other»-term a number of times, which to me appears uninspired and unecessary. Consider revising the Introduction section in particular and see if this could be shortened and more stringent, make it less «text-book» and more scientific paper.

I have a few comments on the use of terms, that the authors may consider revising. These are suggested minor revisions. However, I have a few comments that I consider in need of a major revision before publication is recommended

Major revisions

For the requested PRISMA-chart the authors provide an empty chart with no relevant information. This must clearly be a mistake. The authors should answer all 22 items in the chart, by indicating on which page the particular information is found. If not all 22 items is described, this should be explained.

The authors did not review any pharmacological interventions from that region, and do not explain sufficiently why they excluded such interventions. Knowing about the challenges in pharmacogenetic variations in Asian subpopulation, one would be interested in whether such variations represent some challenges in the treatment of elderly in South Asia. Check for instance: https://doi.org/10.1111/cts.12771

The authors do not discuss whether they think they lost some crucial information regarding interventions when they decided only to include English language papers. They should do that.

All 19 interventions proved to be effective, which clearly must lead to some discussion on publication bias – although the authors state that this is not part of a scoping review. I think it should be addressed anyway, not as a systematic review of the risk of bias in the retrieved articles, but simply as a few comments on the problems that presenting these results encounter. For instance, could the interventions in itself be effective merely because they were interventions, as opposed to no intervention or program at all?

The authors state:

• No study representation from six South Asian countries highlighted research disparity and knowledge gap in this region, which may associate with a lack of acknowledging geriatric mental health as a major health burden, leaving older adults at mental health risk.

o To me this appear as a speculation. This topic is important in itself, and the authors should dwelve more thoroughly on it; why is reporting from these countries scarce? Are there other possible explanations? Poverty, lack of research funding in general?

Minor revisions

• In abstract:

o caused by declining fertility and increasing life expectancy.

Consider using the term «fertility rates»

• Introduction

o Growing elderly population is an imminent concern

The growing

o A drastic drop in fertility and increasing life expectancy during the late 20th and 21st centuries has eventually

Fertility rates. Consider using plural «have»

o shifting towards consistently burgeoning older adults

reconsider: consistently burgeoning group of older



• The evidence was predominantly based on India, whereas only three articles were identified from Pakistan

o Consider omitting «only»

• Although evidence about geriatric mental health disorders in South Asia is highly scarce, continually growing older adults creates a high demand for understanding their complex physical, mental, functional, and other psychosocial problems and associated health and social needs. Mental health deeply encompasses all aspects of human lives, including social, cultural, religious, spiritual, historical, and holistic, which may incur stressors of varying extents

o Reconsider whether this section in necessary

• Results:

o recruited asample population

please correct

o Although limited, variousinterventions

Please correct

7. PLOS authors have the option to publish the peer review history of their article (what does this mean?). If published, this will include your full peer review and any attached files.

Reviewer #1: **Yes: **Carmen Maria da Silva Maciel Andrade

Reviewer #2: No

Reviewer #3: **Yes: **Eivind Aakhus

---

## [Author Response · Author response to Decision Letter 1]

24 May 2023

Respond to reviewer’s comments

Reviewer’s comment Response

1. Avoid using nonstandard abbreviations such as "MH".

Thanks to the reviewer for this comment. We revised this part in abstract accordingly. 

2. "... limited, variousinterventions" - typo here.

Thank you so much. We have revised accordingly. 

3. The authors seem to allude to the supposed effectiveness of digital or tech-based interventions and lament at the paucity of research on such interventions in South Asia. If so, what are the data and findings on effectiveness? More elaboration can be provided in the discussion section.

Thank you so much for this valuable comment. We raised the issue regarding South Asia's lack of digital or tech-based mental health intervention. Hence, we found only two articles that applied tech-based interventions among older adults with cognitive impairment. We provided a summary and further explanation of the evidence in the result section (5th paragraph – 3.2 Mental health interventions and their outcomes) and discussion section (4th paragraph- Discussion). 

Reviewer 1

Title: 

- There seem to be two titles:

(i) "Mental Health Interventions for Older People in South Asia: A Scoping Review";

(ii) "Effectiveness of mental health interventions for older adults in South Asia: A scoping review". The second title is more complete and better reflects the content of the article.

Thanks for this comment. The initial title of this manuscript was "Mental Health Interventions for Older People in South Asia: A Scoping Review"; however, during the first round of revision, we changed our title to "Effectiveness of mental health interventions for older adults in South Asia: A scoping review" since reviewers of first round suggested so.

- The keywords reflect the content; however, I suggest you include "effectiveness" or "outcomes" or "impact" or "health gains".

Thanks to the reviewer for this comment. We included the keywords suggested by the reviewer. 

Introduction:

- It's clear, short and simple.

- The introduction reveals an adequate understanding of the relevant literature in the field and cites an appropriate range of literature sources. 

Thanks to the reviewer. I appreciate this feedback. 

3 - Methodology:

The methods employed are adequate.

However:

- A supplementary file with the PRISMA-ScR should be submitted.

Thanks to the reviewer for appreciating our work. We have submitted a completed PRISMA-ScR in supplementary section (S1 checklist).

- Quality/risk of bias assessment of included studies is missing 

In this review, we did not conduct any methodological quality/ risk of bias assessment of included studies. According to the JBI manual, Scoping review can map the available evidence and identify/ clarify key concepts in a given field (1,2). It is useful to undertake as a preliminary exercise or to examine evidence when a clear knowledge gap exists (1–3). Since scoping review intends to map existing evidence to a broader extent rather than synthesizing clinically meaningful information. Therefore, critical appraisal or risk of bias assessment is not recommended in the existing scoping review guidelines (3,4), and we mentioned it in our study limitation (in the last paragraph of the discussion section).

4 - Results:

The results are clearly presented and properly analyzed.

However, there is the inclusion of a study with an intervention that can be considered a pharmacological intervention; however, there is an eligibility criterion that excludes studies with pharmacological interventions. I suggest that the results and conclusions clearly relate to the review question, namely the conceptualization and measurement of well-being. As this is a scope review, I suggest that the gap found and the measures to be overcome be presented more clearly and explicitly.

Thanks to the reviewer for this comment. This review included articles that considered any non-pharmacological or psychosocial mental health interventions among South Asian older adults. Hence, if a psychosocial intervention was applied in combination with any pharmacological treatment or assigned pharmacological medication as a comparator to evaluate the comparative effectiveness of psychosocial interventions was considered eligible. That’s why we included those interventions which offered yoga or other meditative interventions in combination with ayurvedic medicine. We have revised the eligibility criteria in the method section to reflect this approach with more clarity. Thanks again. 

Reviewer 2 

General comment: 

First of all, the manuscript is well written, and English is mostly excellent in my opinion.

I would have preferred a less «wordy» approach. To me some sequences, both in the introduction and the discussion sections appear as the authors is rattling off a list of examples, that not necessarily need to be listed. For instance the authors use the «and other»-term a number of times, which to me appears uninspired and unecessary. Consider revising the Introduction section in particular and see if this could be shortened and more stringent, make it less «text-book» and more scientific paper.

I have a few comments on the use of terms, that the authors may consider revising. These are suggested minor revisions. However, I have a few comments that I consider in need of a major revision before publication is recommended.

Thanks to the reviewer for this comment. We revised the introduction part to make it shortened. We wanted to provide a clear picture of geriatric mental health problems in South Asia with its continually growing population aging. We described how countries in this region had been affected by various mental disorders due to different socio-economic, cultural, or systematic/ structural challenges. We highlighted how critical it is to address the intersection of geriatric mental and physical health problems due to the lack of strategic plans, programs, and implementations, which have created a serious knowledge gap and hindered the implementation of contextually appropriate mental health interventions to meet their healthcare and social support. 

Major revisions

For the requested PRISMA-chart the authors provide an empty chart with no relevant information. This must clearly be a mistake. The authors should answer all 22 items in the chart, by indicating on which page the particular information is found. If not all 22 items is described, this should be explained. 

Thank you for this important comment. We included a completed PRISMA-chart with all the required information in the supplementary section (S1 Checklist ). 

The authors did not review any pharmacological interventions from that region, and do not explain sufficiently why they excluded such interventions. Knowing about the challenges in pharmacogenetic variations in Asian subpopulation, one would be interested in whether such variations represent some challenges in the treatment of elderly in South Asia. Check for instance: https://doi.org/10.1111/cts.12771.

Thanks to the review for this valuable comment. We reviewed our paper and discussed with the authors how to communicate better our study objectives and findings. Then we rephrased the background (last paragraph – Introduction) and method (Point c – Eligibility criteria in method section) to accurately rephrase our emphasis on non-pharmacological interventions. Of note, we recognize the importance of pharmacological interventions, but since this paper is conceptualized and conducted from community or population-level mental health perspectives. Therefore, we included community or population-based non-pharmacological or psychosocial interventions such as behavioral or learning-based therapy, group-based therapy, occupational therapy, etc., among South Asian older adults. Again, we highly appreciate your comment. In our limitation, we again mentioned that this review focused on non-pharmacological interventions; however, future research should consider pharmacological and non-pharmacological evidence to make informed decisions for mental health practice (Discussion- 9th paragraph). 

The authors do not discuss whether they think they lost some crucial information regarding interventions when they decided only to include English language papers. They should do that.

Thank you for this comment. Yes, we agreed that this is a limitation of our review, and we stated this limitation in the limitations section (Discussion-9th paragraph). 

All 19 interventions proved to be effective, which clearly must lead to some discussion on publication bias – although the authors state that this is not part of a scoping review. I think it should be addressed anyway, not as a systematic review of the risk of bias in the retrieved articles, but simply as a few comments on the problems that presenting these results encounter. For instance, could the interventions in itself be effective merely because they were interventions, as opposed to no intervention or program at all?

 Thank you so much for your valuable comment. We agreed that there might be some risk of potential publication bias. Therefore, we further elaborated on this issue along with the potential problems of study design and their implications in the discussion section (Discussion – 2nd paragraph).

No study representation from six South Asian countries highlighted research disparity and knowledge gap in this region, which may associate with a lack of acknowledging geriatric mental health as a major health burden, leaving older adults at mental health risk.

o To me this appear as a speculation. This topic is important in itself, and the authors should delve more thoroughly on it; why is reporting from these countries scarce? Are there other possible explanations? Poverty, lack of research funding in general?

Thank you so much for bringing this to our notice. We discussed this in the result section (Result- last paragraph). However, to bring more clarity, we have rephrased this sentence to highlight possible underlying factors of the lack of research representation in this field in South Asia (Discussion-8th Paragraph).

• In abstract:

o caused by declining fertility and increasing life expectancy.

Consider using the term «fertility rates» 

Thank you so much. We have revised our manuscript accordingly (Objective section in Abstract). 

• Introduction

o Growing elderly population is an imminent concern

The growing 

Thank you so much. We have revised our manuscript accordingly (3rd sentence in the introduction first paragraph).

o A drastic drop in fertility and increasing life expectancy during the late 20th and 21st centuries has eventually

Fertility rates. Consider using plural «have» 

Thank you very much. We have revised our manuscript accordingly (5th sentence in the introduction first paragraph).

o shifting towards consistently burgeoning older adults

reconsider: consistently burgeoning group of older 

Thank you for this comment. We have revised our manuscript accordingly (7th sentence in the introduction first paragraph).

• The evidence was predominantly based on India, whereas only three articles were identified from Pakistan

o Consider omitting «only» 

Thank you very much. We have revised our manuscript accordingly (6th sentence of result section in the abstract).

• Although evidence about geriatric mental health disorders in South Asia is highly scarce, continually growing older adults creates a high demand for understanding their complex physical, mental, functional, and other psychosocial problems and associated health and social needs. Mental health deeply encompasses all aspects of human lives, including social, cultural, religious, spiritual, historical, and holistic, which may incur stressors of varying extents

o Reconsider whether this section in necessary. 

Thank you so much for this comment. We rephrased this part in the revised manuscript. 

• Results:

o recruited asample population

please correct

Thanks for this comment. We have revised our manuscript accordingly.

o Although limited, variousinterventions

Please correct

Thanks to the reviewer for this comment. We have revised our manuscript accordingly.

Reviewer 3 

According to Sieber (2007), medical treatment of the elderly (geriatrics) starts from the age of 65 years old (https://pubmed.ncbi.nlm.nih.gov/17934704/). Could you please explain why relevant research in South Asia includes adults from the age of 50 years old? Do the terms “older adults” and “elderly” refer to the same age range? 

Thanks to the reviewer for bringing this to our notice. There is a lack of consensus on the age definition of geriatric people in South Asia. Also, in the population sub-group, life expectancy can differ significantly. Hence, the Western definition of aging, such as age 65+, may not be applicable to South Asian countries, where overall life expectancies have increased in recent decades. Life expectancy in South Asian countries was between 50 and 60 years old up to 2000, while it increased to 60 or 70 years in 2020. Balachandran et al. (2020), hence mentioned that the old-age threshold was lower than 65 years in most parts of Asia in 2012 due to their very low life expectancy. Hence, it is notable that a country with a high mortality rate at young and adult ages, reaching the old-age threshold, will be more unlikely than a country with lower mortality (5). The exceptionality of reaching a particular age in a country will determine the status of the elderly (6,7). So, a person in a country who reaches the age at which only a small percentage of the population reaches is likely to be considered older than someone who reaches that age in a country where this is quite common (5). Recent studies in South Asia started using “older age” or “elderly” at 60 years due to a steady trend of increasing life expectancy in the last few decades; however, based on social context, aging starts even earlier than 60. In this review, we found three articles that included older adults, starting from age 50 or 55 years and above (8–10), but the mean age of all these studies was over 60 years. Therefore, our overall evidence does not shift to the 50-age group. In this review, we included primary studies that defined the aging or geriatric population from their perspective rather than ours. 

Pg. 4: “Mental health deeply encompasses all aspects of human lives, including social, cultural, religious, spiritual, historical, and holistic, which may incur stressors of varying extents.” What exactly do you mean by the term “holistic”? It seems to be a pleonasm, unless it has a different meaning from “all aspects of human lives”. 

Thank you so much for this important comment. We agreed that using “holistic” in this sentence is redundant. Therefore, we removed this word. 

Pg. 4: “These stressors can result in social alienation, loneliness, and psychosocial distress among older adults that may eventually lead to various mental and behavioral disorders, thereby impacting the overall geriatric quality of life.” Could you please add a reference? 

Thanks to the reviewer. We have included references for this statement. 

Pg. 5: “In addition, socioeconomic and cultural dimensions of growing geriatric mental health problems warrant different psychosocial and community-level interventions that are contextually appropriate to develop an older-friendly environment and social support network.” Could you please provide data supporting this statement, either from South Asia or from other regions or continents? 

Thank you for this valuable comment. Socioeconomic and cultural factors are considered important components of individual lifestyle, health-related behavior, and social interaction patterns. Since physical and mental well-being is interconnected, lifestyle factors were identified as proximate determinants to influence elderly mental health (11). This study found direct and indirect associations of socioeconomic or lifestyle factors with elderly mental health in India (11). We have added a reference for the referred statement. Thanks again.

Pg. 5: “To the best of our knowledge, there is a lack of evidence-based reviews exploring mental health interventions among South Asian older adults.” 

Are there relevant reviews for other regions or continents? If so, could you please provide their findings, in order to compare them with yours in Discussion and enrich your suggestions? 

Thank you for this comment. We found many review papers in other regional contexts regarding elderly mental health. However, most studies were focused on specific mental disorders instead of a comprehensive mapping like ours in the South Asian context. Therefore, in our discussion, we have included references, where needed, relevant to our discussion. 

Pg. 5: “For scholarly data, we systematically searched Medline, American Psychological Association (APA) PsycInfo, Academic Search Ultimate, Cumulative Index to Nursing and Allied Health Literature (CINAHL), Health Policy Reference Center, and the Web of Science databases from the date of inception of each database up to August 5, 2022, using the specific set of keywords applied with Boolean operators (i.e., “OR,” “AND”)”. Could you please specify the “date of inception”?

Thank you for this comment. Regarding your comment, the inception date means when the databases were active. For example, Medline remains one of the oldest health sciences databases, whereas PsycInfo or Web of Sciences appeared much after Medline. Therefore, including databases from the beginning of their operationalization ensures that we have included all titles since those databases became active. 

Pg. 5-6: “The search queries were used across titles, abstracts, subject-specific keywords, and topics fields in respective databases.” Could you please specify if you set any filter regarding the date of publication etc.?

Thank you for this comment. In our study, we went through an inclusive search process and did not apply any filter option. We mentioned it in the method section (2.1 Data sources and guidelines – Materials and methods). 

Table 1: Is the search strategy and the keywords presented exactly the same for all the databases that were used? If not, I suggest that you provide an abstract description of the terms that you used in the main text of Methodology and present the database you refer to as an example in Table 1. 

Thank you for your concern. We used the same search strategy (shown in Table 1) for all databases. Therefore, we did not provide any brief description of such a long search strategy in the abstract. Rather, we have included a supplementary document depicting the detailed search process in selected databases (S2 file).

Table 1: Why did you choose “cognitive behavio*” as a keyword? For example, is there evidence that CBT is more effective or usual than other psychotherapeutic approaches in this field? If so, please add the respective information in the Introduction. Similarly, in the Introduction you could add information on every disorder you chose to include in your keywords (e.g., anorexia, schizophrenia), always regarding older adults. 

Thank you very much for your comment. Rather than comparing cognitive behavioral vs. non-cognitive behavioral interventions, we included this keyword because many mental health intervention reviews and meta-analyses included studies with cognitive behavioral therapies (12–14). Cognitive behavioral therapy is widely recognized as one of the many evidence-based interventions by the World Health Organization and the American Psychiatric Association. Therefore, we added this keyword so that if any relevant studies could appear from our search to make our scoping review more inclusive. Also, since mental health disorders encompass a broader area involving changes in emotion, thinking or behavior, or a combination of these, we included mental disorders and psychological problems within the scope of the International Classification of Diseases (ICD) or Diagnostic and Statistical Manual (DSM) for mental disorders. We have mentioned this in our study eligibility criteria (2.2 Eligibility criteria- Method section). 

Eligibility criteria: Please, specify if you included both quantitative and qualitative research, as well as case studies/practice papers, or if you set any limitations regarding method. 

Thank you so much for this comment. We have specified in our eligibility criteria what type of study we were looking for. We searched for studies that applied any intervention for older adults and reported their effects on mental health outcomes. Though intervention studies are more likely to be quantitative; however, we did not limit our criteria in terms of quantitative or qualitative research. 

Eligibility criteria: Please, specify if there were any limitations regarding sex, gender, ethnicity, physical/mental state, or other demographics that you may consider important. 

Thank you for this comment. We have described the eligibility criteria of this review paper in the Method section. There are no more criteria other than that.

Eligibility criteria: “Studies that presented one or more psychosocial, behavioral, or non-pharmacological intervention/s for geriatric mental health problems were included in this review.” Could you please provide the definitions of "psychosocial therapy/ intervention" and "behavioral intervention" in the Introduction and explain how these apply to older adults (e.g., their main focus or components)?

Thanks to the reviewer for bringing this important issue to our notice. We have explained the definition of psychosocial interventions, including their potential implications at the population level (Introduction - 4th paragraph).

Data extraction: “We used a pre-designed data extraction form that included items on study characteristics…” Was the form designed by the authors? If so, could you please clarify why you did not prefer a standardized form like PICO or SPIDER?

Thank you very much for this comment. PICO was considered for operationalizing the study, such as the development of the search strategy, screening of individual studies, and finally, the data extraction form was informed by previous scoping reviews consisting of key variables of interest. Those key variables, all together, reflected PICO and other relevant information from each sub-component, for example, I = Intervention in PICO; hence we included intervention/ study design, including sampling strategy, timeframe, and sample size in each intervention group (Table 2). Moreover, O= outcome in PICO, and we provided a separate column for outcomes in Table 2.

Characteristics of the included studies: Please provide information regarding the sex/gender of the participants, as well as their physical/mental/emotional state if mentioned in the studies. 

Thank you for this comment. We described all the important characteristics of included studies in Table 2. We also have highlighted the important study characteristics in the result section (3.1 Characteristics of the included studies- Result section). Table 2 mainly provides a summary of each study included in this review regarding study design, study participants, settings, and sample characteristics. In this revised manuscript, we added information regarding the sex/gender of the participants of individual studies.

Characteristics of the included studies: Please, specify if there were any Randomized Controlled Trials. 

Thank you for bringing this to our notice. We have added this in the result section (3.1 Characteristics of the included studies- Result section). Also, the methodology of individual studies was described in the table (Table 2). 

Table 2: Please, note the statistical significance of the findings, where appropriate. ("*" for p.05, "**" for p.01, "***" for p.001). 

- Table 2: Please, explain the abbreviations below the Table in the form of an annotation. 

- Table 2: A briefer description of the data would be preferable. Perhaps, you could add further details about the studies in the main text. Also, arranging the studies based on their method and/or therapeutic approach would be helpful for the reader. 

Thank you so much for bringing this to our notice. We have revised the table (Table 2) and included the p-value for all findings to show the statistical significance. We have added the main study outcomes following interventions, including other important components such as study design, sample participants, and applied interventions. Therefore, it was not possible to shorten the table description. Moreover, a description of the study in Table 2 may be helpful for the reader to get overall information about that study. Thank you again.

Results: It would be better if you presented the interventions along with their outcomes in separate subsections based on their approach. 

Thank you for this comment. Although few studies were included in this review; however, varied interventions or approaches were applied to target different mental disorders. Hence, we broadly categorized them based on their intervention approaches. However, categorizing based on outcome was impossible since studies mostly evaluated interventions on multiple outcomes. Since our objective of this review was to explore and summarize available evidence about mental health interventions for geriatric mental disorders; therefore, we discussed them mainly based on assigned interventions. Moreover, we have discussed the outcomes and interventions in a separate part of the result section (3.2 Mental health intervention and outcomes- Result section).

Results: Although not necessary for a scoping review, it would be interesting if you assessed the quality of the studies, based on established criteria for Empirically Supported Theapies (e.g., Chambless et al., 1998), and present their limitations. In that way, it would be easier to identify methodological gaps and make fruitful suggestions for future studies.

Thanks to the reviewer for this comment. In this review, we did not conduct any methodological quality/ risk of bias assessment of included studies. According to the JBI manual, Scoping review can map the available evidence and identify/ clarify key concepts in a given field (1,2). It is useful to undertake as a preliminary exercise or to examine evidence when a clear knowledge gap exists (1–3). Since scoping review intends to map existing evidence to a broader extent rather than synthesizing clinically meaningful information, therefore critical appraisal or risk of bias assessment is not recommended in the existing scoping review guidelines (3,4), and we mentioned it in our study limitation (in the last paragraph of the discussion section). However, we discussed the potential challenges regarding study design and methodology in the discussion section (2nd paragraph and 8th paragraph- Discussion section) 

Discussion: “Scoping reviews are useful for mapping information from existing literature on a certain topic to a broader extent, including research findings and gaps.” 

This sentence is not necessary. 

Thank you for this comment. We kept this sentence to inform the broader audience. However, since the reviewer believes it is irrelevant, we removed this sentence. 

Discussion: “Yoga, Tai chi, and other meditative movements have historically been practiced in South Asia, predominantly assigned as an intervention to explore their effectiveness on geriatric mental disorders.” 

Could you please provide a reference?

Thanks to the reviewer for this comment. We have included reference for this statement. 

Discussion: “Exploring and summarizing evidence…which were likely to increase the risk of bias.” Again, it is suggested that methodology is evaluated more systematically through established criteria for Empirically Supported Therapies (see for example Chambless et al., 1998).

Thank you for this comment. According to JBI guidelines, scoping review intends to map existing evidence to a broader extent rather than synthesizing clinically meaningful information. Therefore, critical appraisal or risk of bias assessment is not recommended in the existing scoping review guidelines (3,4). In this review, we explored to have broader knowledge or understanding regarding mental health interventions among the geriatric population; therefore, we followed JBI guidelines and did not perform study quality assessments. But we did discuss many potential challenges we identified in the studies included. Hence, we encourage future researchers to adopt more robust methodological approaches in designing intervention studies on geriatric mental health (8th paragraph- Discussion section). 

Discussion: “These interventions mainly focused on geriatric mental health outcomes, where associated physical, psychological, socioeconomic, and cultural factors remained largely unspecified.” Is there relevant literature - even from other regions of the world - that highlights the importance of these factors in geriatric mental health care?

Thank you for this comment. Many studies have mentioned geriatric physical, socioeconomic, and cultural aspects as important factors associated with mental health problems. For example, older adults in low socioeconomic conditions having chronic morbidities are more likely to develop mental disorders (15–17). In addition, cultural beliefs are also associated with elderly mental health treatment preferences and healthcare decisions (18). 

Discussion: “However, it has been evident that mental health care utilization among older people 

is low (59–61); hence, assessing and improving geriatric mental health, especially in low- and middle income countries, including South Asia, becomes critical (62).” 

Is there any evidence or assumption on why mental health care utilization among older people is low? How exactly can assessment and improvement of geriatric mental health solve this issue? Any ideas from the studies included in your research? 

Thank you for this comment. We provided references to our statement in the discussion. Many countries worldwide have found low healthcare utilization among older adults, which mostly depends on various contextual factors. In this review, studies evaluated the effect of various mental health interventions among older adults. Hence, discussion/ assumption about low geriatric health care utilization probably was beyond their scope. However, since low healthcare utilization is closely oriented to the country’s healthcare investment, healthcare structure, policies and strategic plan of health systems, political agenda, socioeconomic or many other cultural factors, therefore, to evaluate elderly low-healthcare utilization, country-specific multidisciplinary research is required. 

Discussion: “The cumulative impact and early-life stressors caused by post-colonial sociocultural reforms in India and Pakistan… and living patterns which compelled older adults to live independently.” Could you please provide references? 

Thank you very much for this comment. Prior studies reflected colonialism and Indian civilization and how historically that impacted the development of psychiatry and population mental health (19–21). Hence, we provided country-specific references supporting our statement (22–25). There has been a long political conflict in this region since British colonialism that developed the world’s greatest refugee crisis, political turmoil, and civil unrest in India, Pakistan, and Bangladesh. Nepal and Sri Lanka also had a long history of internal conflict. Such conflict and war can cause a profound mental health impact, especially among children, women, adolescents, and elderly people (26). In most cases, war-torn countries do not invest in mental health promotion because of a lack of resources, understanding, mismanagement, or corruption (26), which can eventually trigger a cumulative mental health impact. 

Discussion: “therefore, we encourage future researchers to adopt more robust methodological approaches in designing intervention studies on geriatric mental health.” 

How do you interpret the methodological limitations? Do you have any suggestions on specific ways of resolving them? Could you compare the different therapeutic approaches in terms of method, outcomes, and effectiveness?

Thank you for this important comment. In the discussion section, we discussed study characteristics, study design, study settings, sampling strategies, intervention approach, and mental health outcomes. We also highlighted methodological limitations in different studies, including their possible impact on results (2nd paragraph -Discussion section); hence, we recommended that future researchers adopt more robust methodological approaches in designing intervention studies on geriatric mental health (8th paragraph- Discussion section).

Proof reading by a native English speaker is recommended. 

Thank you for this feedback. A native English speaker has revised this revised version of manuscript.

References: 

1. 11.1.1 Why a scoping review? - JBI Manual for Evidence Synthesis - JBI Global Wiki [Internet]. [cited 2023 Jan 24]. Available from: https://jbi-global-wiki.refined.site/space/MANUAL/4687794/11.1.1+Why+a+scoping+review%3F

2. Arksey H, O’Malley L. Scoping studies: towards a methodological framework. Int J Soc Res Methodol [Internet]. 2005 Feb 1 [cited 2023 Jan 24];8(1):19–32. Available from: https://doi.org/10.1080/1364557032000119616

3. Peters MDJ, Godfrey CM, Khalil H, McInerney P, Parker D, Soares CB. Guidance for conducting systematic scoping reviews. JBI Evid Implement [Internet]. 2015 Sep [cited 2023 Jan 24];13(3):141. Available from: https://journals.lww.com/ijebh/FullText/2015/09000/Guidance_for_conducting_systematic_scoping_reviews.5.aspx

4. Peters MDJ, Marnie C, Tricco AC, Pollock D, Munn Z, Alexander L, et al. Updated methodological guidance for the conduct of scoping reviews. JBI Evid Synth. 2020 Oct;18(10):2119–26. 

5. Balachandran A, de Beer J, James KS, van Wissen L, Janssen F. Comparison of population aging in Europe and Asia using a time-consistent and comparative aging measure. J Aging Health. 2020;32(5–6):340–51. 

6. Angus J, Reeve P. Ageism: A threat to “aging well” in the 21st century. J Appl Gerontol. 2006;25(2):137–52. 

7. Giles H, Reid SA. Ageism Across the Lifespan: Towards a Self-Categorization Model of Ageing. J Soc Issues [Internet]. 2005 [cited 2023 May 20];61(2):389–404. Available from: https://onlinelibrary.wiley.com/doi/abs/10.1111/j.1540-4560.2005.00412.x

8. Arshad H, Anwar K, Khattak H, Amjad I, Majeed Y. Effect of Brain Training Game on Mild Cognitive Impairment (MCI) in Older Adults. PJMHS; 2021. 

9. Chobe S, Patra SK, Chobe M, Metri K. Efficacy of Integrated Yoga and Ayurveda Rasayana on cognitive functions in elderly with mild cognitive impairment: Non-randomized three-arm clinical trial. J Ayurveda Integr Med [Internet]. 2022 Jan 1 [cited 2023 May 20];13(1):100373. Available from: https://www.sciencedirect.com/science/article/pii/S0975947620301066

10. Chobe SV, Chobe M, Metri K, Raghuram N. Yoga and Rasayana for Mental Health in Age-related Cognitive Decline - A Controlled Trial. Indian J Sci Technol [Internet]. 2022 Mar 5 [cited 2023 May 20];15(9):364–70. Available from: https://indjst.org/

11. Bhandari P, Paswan B. Lifestyle Behaviours and Mental Health Outcomes of Elderly: Modification of Socio-Economic and Physical Health Effects. Ageing Int [Internet]. 2021 Mar 1 [cited 2023 May 21];46(1):35–69. Available from: https://doi.org/10.1007/s12126-020-09371-0

12. Huntley AL, Araya R, Salisbury C. Group psychological therapies for depression in the community: systematic review and meta-analysis. Br J Psychiatry [Internet]. 2012 Mar [cited 2023 May 23];200(3):184–90. Available from: https://www.cambridge.org/core/journals/the-british-journal-of-psychiatry/article/group-psychological-therapies-for-depression-in-the-community-systematic-review-and-metaanalysis/B61A8196EB81693A420A5ABB52EA4C93

13. McQuaid JR, Granholm E, McClure FS, Roepke S, Pedrelli P, Patterson TL, et al. Development of an Integrated Cognitive-Behavioral and Social Skills Training Intervention for Older Patients With Schizophrenia. J Psychother Pract Res [Internet]. 2000 [cited 2023 May 23];9(3):149–56. Available from: https://www.ncbi.nlm.nih.gov/pmc/articles/PMC3330598/

14. Hofmann SG, Asnaani A, Vonk IJJ, Sawyer AT, Fang A. The Efficacy of Cognitive Behavioral Therapy: A Review of Meta-analyses. Cogn Ther Res [Internet]. 2012 Oct 1 [cited 2023 May 23];36(5):427–40. Available from: https://doi.org/10.1007/s10608-012-9476-1

15. Bøen H, Dalgard OS, Bjertness E. The importance of social support in the associations between psychological distress and somatic health problems and socio-economic factors among older adults living at home: a cross sectional study. BMC Geriatr [Internet]. 2012 Jun 8 [cited 2023 May 21];12(1):27. Available from: https://doi.org/10.1186/1471-2318-12-27

16. Chou KL, Chi I. Prevalence of depression among elderly Chinese with diabetes. Int J Geriatr Psychiatry [Internet]. 2005 [cited 2023 May 21];20(6):570–5. Available from: https://onlinelibrary.wiley.com/doi/abs/10.1002/gps.1328

17. Zhang Y, Chen Y, Ma L. Depression and cardiovascular disease in elderly: Current understanding. J Clin Neurosci [Internet]. 2018 Jan 1 [cited 2023 May 21];47:1–5. Available from: https://www.sciencedirect.com/science/article/pii/S0967586816306282

18. Jimenez DE, Bartels SJ, Cardenas V, Dhaliwal SS, Alegría M. Cultural Beliefs and Mental Health Treatment Preferences of Ethnically Diverse Older Adult Consumers in Primary Care. Am J Geriatr Psychiatry [Internet]. 2012 Jun 1 [cited 2023 May 21];20(6):533–42. Available from: https://www.sciencedirect.com/science/article/pii/S1064748112620673

19. How Colonialism and the British Monarchy Impacted Sustainable Development Mental Health [Internet]. SustainabilityX. 2023 [cited 2023 May 24]. Available from: https://www.sustainabilityx.co/post/how-colonialism-and-the-british-monarchy-impacted-sustainable-development-mental-health

20. Sharma S. Psychiatry, colonialism and Indian civilization: A historical appraisal. Indian J Psychiatry [Internet]. 2006 [cited 2023 May 23];48(2):109–12. Available from: https://www.ncbi.nlm.nih.gov/pmc/articles/PMC2913558/

21. Mishra A, Mathai T, Ram D. History of psychiatry: An Indian perspective. Ind Psychiatry J. 2018;27(1):21–6. 

22. Luitel NP, Jordans MJD, Sapkota RP, Tol WA, Kohrt BA, Thapa SB, et al. Conflict and mental health: a cross-sectional epidemiological study in Nepal. Soc Psychiatry Psychiatr Epidemiol [Internet]. 2013 Feb 1 [cited 2023 May 21];48(2):183–93. Available from: https://doi.org/10.1007/s00127-012-0539-0

23. Basnet S, Kandel P, Lamichhane P. Depression and anxiety among war-widows of Nepal: a post-civil war cross-sectional study. Psychol Health Med [Internet]. 2018 Feb 7 [cited 2023 May 21];23(2):141–53. Available from: https://doi.org/10.1080/13548506.2017.1338735

24. Chandradasa M, Kuruppuarachchi K a. LA. Child and youth mental health in post-war Sri Lanka. BJPsych Int [Internet]. 2017 May [cited 2023 May 21];14(2):36–7. Available from: https://www.cambridge.org/core/journals/bjpsych-international/article/child-and-youth-mental-health-in-postwar-sri-lanka/0882A478DDB02FC355718F77993F1461

25. Alam F, Hossain R, Ahmed HU, Alam MT, Sarkar M, Halbreich U. Stressors and mental health in Bangladesh: current situation and future hopes. BJPsych Int [Internet]. [cited 2023 May 24];18(4):91–4. Available from: https://www.ncbi.nlm.nih.gov/pmc/articles/PMC8554941/

26. Abi-Rached JM. Post-War Mental Health, Wealth, and Justice. Traumatology [Internet]. 2009 Dec 1 [cited 2023 May 21];15(4):55–64. Available from: https://journals.sagepub.com/doi/abs/10.1177/1534765609359730

---

## [Decision Letter · Decision Letter 2]

13 Jun 2023

PONE-D-22-29794R2Effectiveness of mental health interventions for older adults in South Asia: A scoping reviewPLOS ONE

Dear Dr. Mazumder,

Thank you for submitting your manuscript to PLOS ONE. After careful consideration, we feel that it has merit but does not fully meet PLOS ONE’s publication criteria as it currently stands. Therefore, we invite you to submit a revised version of the manuscript that addresses the points raised during the review process.

We look forward to receiving your revised manuscript.

Kind regards,

Qin Xiang Ng, MD, MPH

Academic Editor

PLOS ONE

Journal Requirements:

Reviewers' comments:

Reviewer's Responses to Questions

**Comments to the Author**

1. If the authors have adequately addressed your comments raised in a previous round of review and you feel that this manuscript is now acceptable for publication, you may indicate that here to bypass the “Comments to the Author” section, enter your conflict of interest statement in the “Confidential to Editor” section, and submit your "Accept" recommendation.

Reviewer #1: All comments have been addressed

Reviewer #2: All comments have been addressed

Reviewer #3: (No Response)

2. Is the manuscript technically sound, and do the data support the conclusions?

Reviewer #1: Yes

Reviewer #2: Yes

Reviewer #3: Yes

3. Has the statistical analysis been performed appropriately and rigorously? 

Reviewer #1: N/A

Reviewer #2: N/A

Reviewer #3: N/A

4. Have the authors made all data underlying the findings in their manuscript fully available?

Reviewer #1: Yes

Reviewer #2: Yes

Reviewer #3: Yes

5. Is the manuscript presented in an intelligible fashion and written in standard English?

Reviewer #1: Yes

Reviewer #2: Yes

Reviewer #3: Yes

6. Review Comments to the Author

Reviewer #1: Thank you for giving me the opportunity to review this manuscript entitled “Effectiveness of mental health interventions for older adults in South Asia: A scoping review”.

In my opinion, it is an interesting manuscript and brings contributions to the body of knowledge on mental health interventions for older adults

Reviewer #2: Ok.

Reviewer #3: I think the authors have addressed my comments in an acceptable way, and, in my opinion, improving the paper. There are a few questions left that the authors should answer:

1. The total number of articles assessed for inclusion in abstract and results section is not similar

2. "The rapid technological advancement in medical sciences since the 19th century has created different avenues in preventive, promotive, and curative healthcare." Do you mean 19th or 20th century?

3. Typo: about the effectiveness on metal health outcomes

7. PLOS authors have the option to publish the peer review history of their article (what does this mean?). If published, this will include your full peer review and any attached files.

Reviewer #1:** **No

Reviewer #2: No

Reviewer #3: No

---

## [Author Response · Author response to Decision Letter 2]

14 Jun 2023

Response to reviewer’s comments

Reviewer #3: I think the authors have addressed my comments in an acceptable way, and, in my opinion, improving the paper. There are a few questions left that the authors should answer:

1. The total number of articles assessed for inclusion in abstract and results section is not similar. 

Thanks to the reviewer for this comment. We revisited our manuscript and found consistency in the abstract and result section. In the abstract, we reported, "From a total of 3432 potential articles retrieved, 19 were included in this review following pre-determined eligibility criteria" (Refer to 1st line of the results in the abstract). In the results section, we reported similarly – "We initially found 26 potential citations for full-text evaluation and finally included 19 articles in this scoping review (31–49)" (Refer to the 5-6 lines in 1st paragraph in Results section). 

2. "The rapid technological advancement in medical sciences since the 19th century has created different avenues in preventive, promotive, and curative healthcare." Do you mean 19th or 20th century?

Thanks to the reviewer for this important comment. We completely agree with you and revised our manuscript accordingly. Technological advancement in medical sciences started evolving in the 20th century. It gradually expanded in different fields of medical sciences, offering direct services to people with mental disorders (First line in 4th paragraph of Discussion section). Thanks again for bringing this to our notice.

3. Typo: about the effectiveness on metal health outcomes

Thanks to the reviewer. We appreciate this comment. We have addressed the typo, which is now mentioned correctly in the revised manuscript (Refer to the last sentence of the 2nd paragraph in the Discussion section).

---

## [Editor Report · Decision Letter 3]

16 Jun 2023

Effectiveness of mental health interventions for older adults in South Asia: A scoping review

PONE-D-22-29794R3

Dear Dr. Mazumder,

We’re pleased to inform you that your manuscript has been judged scientifically suitable for publication and will be formally accepted for publication once it meets all outstanding technical requirements.

Kind regards,

Qin Xiang Ng, MD, MPH

Academic Editor

PLOS ONE
---

## [Editor Report · Acceptance letter]

28 Jun 2023

PONE-D-22-29794R3 

 Effectiveness of mental health interventions for older adults in South Asia: A scoping review 

Dear Dr. Mazumder:

I'm pleased to inform you that your manuscript has been deemed suitable for publication in PLOS ONE. Congratulations! Your manuscript is now with our production department. 

Kind regards, 

on behalf of

Dr. Qin Xiang Ng 

Academic Editor

PLOS ONE